# Self-supervised Sparse Vision Concepts for Image Understanding and Reconstruction

## Abstract

Self-supervised vision encoders have become critical components of modern machine learning systems. Despite remarkable advances in image understanding, generation, and multimodal alignment, the underlying representation of visual features has remained largely unchanged, constrained by historical architectures and benchmarks. This reliance on dense feature grids introduces redundancy and limits the integration of understanding and generation. We propose a novel framework that represents images with a small number of sparse tokens in the form of low-rank matrix factorization. While mathematically simple, this formulation effectively disentangles semantic and spatial information. We demonstrate that vision-only self-supervised learning under this framework yields sparse token representations that simultaneously support high-quality image understanding, detailed pixel-level reconstruction, and fine-grained semantic understanding. Together, these results highlight sparse tokens as a promising alternative to dense grids for efficient and versatile visual representation learning.

## 1 Introduction

Learning visual representations has been a central pursuit in computer vision since the advent of deep learning (Bengio et al., 2013). Modern vision models transform raw pixels into latent features that power nearly all downstream applications. Architectures have evolved dramatically—from early convolutional networks (Lecun et al., 1998) to ResNets (He et al., 2016) and, more recently, vision transformers (ViTs) (Dosovitskiy et al., 2020). Despite these advances, the geometric format of visual representation has remained largely unchanged: a dense 2D grid of high-dimensional features, each tied to a local patch of the image. This design is natural, since pixels are arranged on a grid.

However, dense representations are highly redundant, as the number of meaningful objects or regions in an image is far smaller than the number of pixels or patches. Prior work has shown that images can be reconstructed from only a small subset of patches (He et al., 2022). Motivated by this redundancy, several methods have already adopted sparse representations in downstream tasks. For instance, Sparse R-CNN (Sun et al., 2021), DETR (Carion et al., 2020), and MaskFormer (Cheng et al., 2021) learn a set of queries from detection or segmentation labels. BLIP-2 (Li et al., 2023) introduced a Q-Former to extract sparse tokens under paired image–text supervision. TiTok (Yu et al., 2024) represents images as sparse 1D tokens for efficient reconstruction and generation, but the semantic quality of these features still lags behind state-of-the-art self-supervised methods.

We take a different path: learning sparse representations directly from images in a fully self-supervised manner (Caron et al., 2021; Oquab et al., 2023; Siméoni et al., 2025; Chen et al., 2020a; Chen & He, 2021; Caron et al., 2020; Grill et al., 2020). Our goal is to obtain a compact set of tokens that support high-quality image understanding and reconstruction at the same time, without human labels or paired data.

To this end, we revisit the fundamentals of visual representation. At its core, perception requires two complementary pieces of information:

1. What objects or visual concepts are present.

2. Where they are located.

The two are unified in forming a holistic representation, but separable in how they behave: the "what" should remain invariant across views, while the "where" changes with viewpoint. Motivated by this principle, we propose to represent an image in the form of low-rank matrix factorization that disentangles these factors. This formulation enables efficient reconstruction from as few as 8 tokens and allows learning the sparse tokens to encode fine-grained semantics in a self-supervised way.

Our contributions can be summarized as follows:

- **Sparse vision representation framework**: We propose STELLAR, an efficient form of latent vision representation modeling an image with only a handful of sparse tokens, by disentangling *what* concepts are present and *where* are they located. The latent representation with as few as 8 tokens can achieve both detailed pixel reconstruction and high-level semantic understanding at the same time.

- **Self-supervised learning method**: We introduce a self-supervised training scheme that learns the sparse latent vision representation without annotation. By discovering and aligning multiple visual concepts across views using optimal transport, we enforces invariance in the "what" while adapting the "where," inducing rich semantic representation.

- **Observations**: (i) STELLAR surpasses prior sparse representation approaches by jointly achieving strong semantic understanding (IN-1k lin. acc. 79.10%) and high-quality reconstruction (FID 2.60). (ii) The sparse image modeling framework induces fine-grained, region-aware semantics even without explicit supervision on the dense feature map, which transfers effectively to downstream tasks with simple linear probing.

## 2 PRELIMINARIES

Representation learning involves encoding an image $X$ with a neural network $\mathcal{E}$ to extract latent features $\boldsymbol{Z} = \mathcal{E}(X)$ for downstream tasks. Traditionally, vision representation takes the dense form

$$\boldsymbol{Z} \in \mathbb{R}^{(h \cdot w) \times d},$$

where $h$ and $w$ denote the height and width of the 2D grid partitioning the image. Each grid location is represented by a feature vector $\boldsymbol{z}_i := \boldsymbol{Z}_{i,:} \in \mathbb{R}^d$ for $1 \leq i \leq h \cdot w$. Many vision models also include a global representation $\boldsymbol{z}_0 \in \mathbb{R}^d$, obtained either by pooling over the feature map or by introducing a [CLS] token in transformers. Even in variational models such as VAEs or VQ-VAEs, where the latent variables $\boldsymbol{z}_i$ are modeled as probability distributions rather than deterministic embeddings, the underlying 2D grid structure remains unchanged.

In contrast, *sparse visual representation* aims to represent the image with

$$\boldsymbol{Z} \in \mathbb{R}^{r \times d}, \quad r \ll h \cdot w.$$

Ideally, the number of sparse tokens $r$ should be less than an order of magnitude than the total number of dense tokens $n = h \cdot w$. In addition, we want the sparse tokens $\boldsymbol{Z}$ to serve as a holistic representation of the image $X$, i.e. $\boldsymbol{Z}$ contains sufficient information to reconstruction the original image, while at the same time possessing rich semantics for downstream tasks. Mathematically, we define such holistic representation as follows:

- **Reconstruction**: There exists a decoder $\mathcal{D}$ such that the sparse features $\boldsymbol{Z} = \mathcal{E}(X)$ can faithfully reconstruct the original image: $\mathcal{D}(\boldsymbol{Z}) \approx X$.

- **Understanding.** For a downstream task with joint distribution $(X, Y) \sim \mathcal{X} \times \mathcal{Y}$ and loss function $\mathcal{L}$, there exists a simple predictor $f \in \mathcal{F}$ such that, using frozen sparse features $\boldsymbol{Z} = \mathcal{E}(X)$, the expected task loss $\mathbb{E}_{(X,Y)}\big[\mathcal{L}(f(\boldsymbol{Z}), Y)\big]$ is low.

While $Y$ can in principle be arbitrary, in downstream tasks it typically reflects human interpretations of the image, such as classification labels, segmentation masks, or textual descriptions. The predictor $f$ is usually drawn from a simple function class $\mathcal{F}$, e.g., a linear layer.

Prior works tend to emphasize only one aspect of "representation." For example, TiTok (Yu et al., 2024) uses 32 tokens to reconstruct an image with 256 patches at high fidelity, but its sparse features

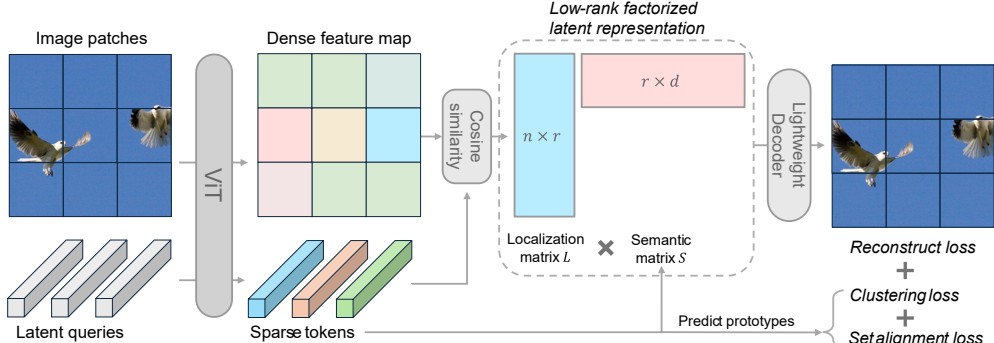

Figure 1: The STELLAR framework. We use a vanilla ViT to extract sparse tokens from an image, and model the latent representation as a low-rank matrix factorization, ensuring reconstruction of the original image. Clustering loss and set alignment loss are applied on the disentangled sparse tokens.

lag far behind other self-supervised vision models in semantic understanding. Conversely, MAE (He et al., 2022) is designed to capture semantics by reconstructing randomly masked patches, yet the resulting reconstructions are often blurry.

This reveals an empirical dilemma in current vision frameworks: models that excel at pixel-level reconstruction often produce weaker semantic representations (Zhang et al., 2022; Chen et al., 2024). Conversely, state-of-the-art SSL methods that achieve strong semantics typically abandon pixel reconstruction to avoid low-level shortcuts (Zhou et al., 2021; Assran et al., 2023; Darcet et al., 2025). In contrast, we demonstrate that by disentangling semantic and localization information, it is possible to learn sparse representations that simultaneously achieve strong image understanding and high-quality reconstruction.

## 3 THE STELLAR FRAMEWORK

### 3.1 SPARSE IMAGE MODELING

Images depict the physical world, which can be understood as a collection of objects located in space. From this perspective, visual information naturally decomposes into two complementary components: (1) *what* objects or concepts are present, and (2) *where* they are located. Unlike dense grid-based representations, which describe what appears at each individual location, we model an image using a compact set of semantic concepts together with their spatial distributions. Concretely, we represent an image with $r$ concept embeddings

$$\boldsymbol{s}_1, \cdots, \boldsymbol{s}_r \in \mathbb{R}^d,$$

where each $\boldsymbol{s}_j$ captures a distinct semantic concept. The spatial distribution of these concepts is expressed through weights $\boldsymbol{l}_1, \cdots, \boldsymbol{l}_n \in \mathbb{R}^r$, where $n$ is the total number of patches. By constraining $0 \leq \boldsymbol{l}_i \leq 1$ and $\boldsymbol{1}^\top \boldsymbol{l}_i = 1$, each patch is represented as a convex combination of the concept embeddings: $\boldsymbol{v}_i = \sum_{j=1}^r \boldsymbol{l}_{i,j} \boldsymbol{s}_j$. Thus, the set $\boldsymbol{s}_j{}_{j=1}^r$ acts as a basis for constructing patch-level features. In matrix form, the latent representation of image $X$ is encoded as:

$$\boldsymbol{S}, \boldsymbol{L} = \mathcal{E}(X), \quad \boldsymbol{S} = [\boldsymbol{s}_1, \ldots, \boldsymbol{s}_r]^\top \in \mathbb{R}^{r \times d}, \quad \boldsymbol{L} = [\boldsymbol{l}_1, \ldots, \boldsymbol{l}_n]^\top \in \mathbb{R}^{n \times r}, \tag{1}$$

which can be combined to form a patch-wise dense representation:

$$\boldsymbol{V} = \boldsymbol{L}\boldsymbol{S}, \quad 0 \leq \boldsymbol{L} \leq 1, \boldsymbol{L}\boldsymbol{1}_r = \boldsymbol{1}_n. \tag{2}$$

Compared to a canonical dense representation of shape $n \times d$, $\boldsymbol{V} = \boldsymbol{L}\boldsymbol{S}$ can be considered as a form of low-rank matrix approximation from the sparse representation. Critically, we enforce that this low-rank approximation can reconstruct the original image through some decoder $\mathcal{D}$:

$$\mathcal{D}(\boldsymbol{L}\boldsymbol{S}) \approx X. \tag{3}$$

This *low-rank approximated reconstruction* ensures that the sparse tokens $\{s_j\}_{j=1}^r$ capture sufficient information to recover the image when combined with their spatial distributions. While the form in equation 2 resembles the low-rank structure used in convex semi-nonnegative matrix factorization (Ding et al., 2008), we do not perform NMF or any matrix factorization algorithm on the feature map (i.e. $LS \approx \mathcal{E}(X)$). Instead, both $S$ and $L$ are learnable latent variables output directly from the forward pass of the encoder, and their product is decoded back to the original image ($\mathcal{D}(LS) \approx X$), allowing an autoencoder-style training. Finally, a compact sparse representation is then obtained by concatenating the concept embeddings with the transposed localization matrix:

$$Z = [S, L^T] \in \mathbb{R}^{r \times d^*}, \quad d^* = d + n. \tag{4}$$

We refer to our framework defined by the form of $S, L = \mathcal{E}(X)$ and $\mathcal{D}(LS) \approx X$ as **S**parse **T**oken **E**xtraction and **L**ocalization with **L**ow-rank **A**pproximated **R**econstruction (STELLAR). We note that the framework is flexible and does not prescribe any specific encoder or decoder architecture. In this work, we adopt a simple design with common modules to obtain $S$ and $L$ as described below.

As illustrated in Fig. 1, the encoder includes a ViT and $r$ learnable latent query vectors, which are passed to the ViT alongside patchified image tokens. Processed by the ViT jointly, the latent queries produce sparse tokens $S \in \mathbb{R}^{r \times d}$, and the image patches output a dense feature map $U \in \mathbb{R}^{n \times d}$.

To obtain the localization matrix $L \in \mathbb{R}^{n \times r}$ associated with the sparse tokens, we project both $S$ and $U$ into a shared embedding space and compute their pairwise cosine similarities, followed by a softmax normalization with temperature $t$:

$$L = \text{softmax}\left(\frac{\text{cossim}(UW_1, SW_2)}{t}\right), \tag{5}$$

where $W_1$ and $W_2$ are learnable linear projections, and $t$ controls the sharpness of the spatial distribution. This mapping is structurally similar to the attention weights obtained in a single-head cross-attention layer, up to the use of L2 normalization and an explicit temperature parameter. We adopt this simple formulation to compute the $L$ matrix, and found it to be stable and effective for learning sparse concept localization across all experiments.

All together, the encoder $\mathcal{E}$ includes a ViT, $r$ learnable latent query vectors, and projection layers $W_1, W_2$. The decoder $\mathcal{D}$ is a lightweight ViT reconstructing the image patches from $LS$.

Although the low-rank form $LS$ is simple, it effectively disentangles high-level semantic concepts from low-level spatial localization. This yields two key benefits for both reconstruction and representation learning:

- The concept matrix $S$ no longer needs to encode spatial information, and can instead focus purely on learning what objects or visual concepts are present. Through the linear combination $LS$, these concepts can be flexibly allocated across spatial locations to form a dense semantic map, enabling efficient reconstruction.

- Because the semantic embeddings in $S$ are independent of location, we can freely apply image transformations while enforcing consistency in the learned concepts. This invariance induces robust high-level semantic features that transfer well to image understanding tasks.

Next we introduce how to learn semantic-rich latent representation as shown in Fig. 2.

## 3.2 Learning Vision Concept Vocabulary

To encourage sparse tokens to represent transferable vision concepts, we structure them into $K$ learnable prototypes $c_1, \cdots, c_K \in \mathbb{R}^p$. A backbone encoder $\mathcal{E}$ maps a mini-batch of $m$ images into sparse features $S^1, \cdots, S^m$. Each token is projected onto the unit sphere $\mathbb{S}^{p-1}$ via a normalized projector $h : \mathbb{R}^d \to \mathbb{S}^{p-1}$, and its similarity to prototypes $C = [c_1, \cdots, c_K]$ gives logits

$$\lambda_j^i = [c_1 \cdot h(s_j^i), \cdots, c_K \cdot h(s_j^i)], \quad i = 1, \ldots, m, \; j = 1, \ldots, r. \tag{6}$$

Soft assignments follow as

$$q_{j,k}^i = \frac{\exp(\lambda_{j,k}^i/\tau)}{\sum_{k'=1}^K \exp(\lambda_{j,k'}^i/\tau)}, \tag{7}$$

Figure 2: Left: Concept clustering and alignment workflow. Right: visualization of learned representation.

where $\tau$ controls sharpness. Direct entropy minimization of $q_j^i$ is unstable due to non-convexity and empty clusters. Following SwAV (Caron et al., 2020) and CAPI Darcet et al. (2025), we compute balanced assignments $\tilde{q}_j^i$ with Sinkhorn-Knopp (temperature $\tilde{\tau} > \tau$), and optimize

$$\mathcal{L}_{\text{cluster}} = \frac{1}{mr} \sum_{i=1}^{m} \sum_{j=1}^{r} \sum_{k=1}^{K} \tilde{q}_{j,k}^i \log q_{j,k}^i. \tag{8}$$

Unlike DINOv2 and SwAV, which use Sinkhorn only for balancing teacher targets, or CAPI, which optimizes prototypes separately, we minimize $\mathcal{L}_{\text{cluster}}$ end-to-end along with other objectives.

### 3.3 SET CONCEPTS ALIGNMENT

STELLAR produces a fixed-size set of sparse tokens invariant to cropping, masking, or resolution. To align features across augmented views without inherent ordering, we use optimal transport as in Fig. 2. Given global view features $s_1, \ldots, s_r$ and partial-view features $s_1', \ldots, s_r'$, the cost matrix

$$\Theta_{j'j} = \|s_{j'}' - s_j\|_2. \tag{9}$$

We solve for an assignment matrix $P$ via entropy-regularized optimal transport:

$$\min_{P \geq 0} \sum_{j',j} P_{j'j} \Theta_{j'j} - \epsilon H(P), \quad P\mathbf{1}_r = P^T \mathbf{1}_r = \frac{1}{r} \mathbf{1}_r, \tag{10}$$

with $H(P) = -\sum j', j P_{j'j} \log P_{j'j}$. We solve for $P$ by Sinkhorn algorithm, and define the matching $\sigma(j') := \arg\max_j P_{j'j}$. We then compute prototype assignments for the partial-view tokens $q_{j'}' = \text{softmax}(C^T h(s_{j'}')/\tau)$, and minimize the set concept alignment loss

$$\mathcal{L}_{\text{align}} = \frac{1}{r} \sum_{j'=1}^{r} \sum_{k=1}^{K} \tilde{q}_{\sigma(j'),k} \log q_{j',k}'. \tag{11}$$

Optionally, the CLS token is treated as another sparse feature with its own projector and prototypes, but not used for reconstruction. In addition, we optionally use an exponential moving average (EMA) updated momentum encoder to encode the target assignments $\tilde{q}$ in equation 8 and equation 11. We observed that using a momentum encoder is essential in the warm-up stage when training from scratch, but suboptimal in subsequent training. We provide detailed results in ablation study. All together, we jointly optimize the following to learn latent representation $S$ and $L$:

- Reconstruction: $\mathcal{L}_{\text{recon}} = \ell(\mathcal{D}(LS), X)$ via a lightweight decoder $\mathcal{D}$.
- Sparse concept clustering: $\mathcal{L}_{\text{cluster}}$ on prototype assignments.
- Set concept alignment: $\mathcal{L}_{\text{align}}$ between global and partial views.
- KoLeo regularization (Sablayrolles et al., 2018) on the sparse tokens from the same image:

$$\mathcal{L}_{KoLeo} = -\frac{1}{r} \sum_{j=1}^{r} \log \left( \frac{1}{2} \min_{j' \neq j} \|\bar{s}_j - \bar{s}_{j'}\|_2 \right), \quad \bar{s}_j := s_j/\|s_j\|.$$

We train the encoder $\mathcal{E}$, decoder $\mathcal{D}$, projector $h$, and prototypes $\boldsymbol{C}$ jointly with the final objective:

$$\min_{\mathcal{E}, \mathcal{D}, h, \boldsymbol{C}} \quad a_1 \mathcal{L}_{\text{recon}} + a_2 \mathcal{L}_{\text{cluster}} + a_3 \mathcal{L}_{\text{align}} + a_4 \mathcal{L}_{\text{KoLeo}}. \tag{12}$$

In summary, we proposed a sparse vision representation $(\boldsymbol{S}, \boldsymbol{L}) = \mathcal{E}(X)$ that explicitly disentangles semantic concepts from their spatial distributions, enabling the latent variables to support both pixel-level reconstruction and high-level semantic understanding. We introduced a simple encoder design to obtain these latent variables and SSL objectives to shape them into transferable visual concepts.

**What is *not* new?** As our focus is not on architectural innovation, we deliberately build on simple, widely used components. The use of learnable latent queries has appeared in Sparse R-CNN (Sun et al., 2021), TiTok (Yu et al., 2024), and many others. The cosine-similarity–softmax mapping is a standard scoring operation that appears in many contexts, including single-head attention (Vaswani, 2017), contrastive learning and InfoNCE objectives (Oord et al., 2018; Chen et al., 2020a).

## 4 RELATED WORK

**Self-supervised Learning** Recent progress in self-supervised visual learning largely focuses on relating global and local views of an image. Contrastive methods such as SimCLR (Chen et al., 2020a) and the MoCo series (He et al., 2020; Chen et al., 2020b; 2021) highlighted the importance of large-scale augmentation and negative sampling, with MoCo introducing a momentum encoder and queue to relax batch-size requirements for efficient training. SwAV (Caron et al., 2020) combined contrastive learning with online clustering, while BYOL (Grill et al., 2020) and SimSiam (Chen & He, 2021) showed that teacher–student or stop-gradient mechanisms suffice without negatives. The DINO family (Caron et al., 2021; Oquab et al., 2023) further revealed emergent segmentation ability by aligning global and cropped views.

In parallel, masked image modeling (MIM) emerged, inspired by masked language modeling. BEiT (Bao et al., 2021) reconstructed discrete tokens from an external tokenizer, while MAE (He et al., 2022) and SimMIM (Xie et al., 2022) simplified the design for pixel reconstruction with asymmetric encoder–decoder. Many followup methods discarded pixel reconstruction and instead performing alignment in the latent space (Zhou et al., 2021; Baevski et al., 2022; Assran et al., 2022; Dong et al., 2022; Chen et al., 2022; Tao et al., 2023; Yang et al., 2024; Liu et al., 2024). Our work differs by exploring *sparse* self-supervised representations, aiming to combine the semantic richness of contrastive/distillation methods with the reconstruction fidelity of MIM, while avoiding their reliance on dense feature grids.

**Sparse Representation** A growing body of work replaces dense feature maps with a compact set of embeddings for downstream tasks. Sparse R-CNN (Sun et al., 2021), DETR (Carion et al., 2020), MaskFormer (Cheng et al., 2021) and Mask2Former (Cheng et al., 2022) learn sparse queries from detection or segmentation supervision. BLIP-2 (Li et al., 2023) introduces a Q-Former to extract sparse tokens for efficient vision–language modeling, trained with paired image–text data. More recently, TiTok (Yu et al., 2024) represents images with sparse 1D tokens for efficient reconstruction and generation. TokenLearner (Ryoo et al., 2021) integrate the sparse token idea into the ViT architecture, improving model efficiency by reducing the number of tokens. SemMAE (Li et al., 2022) uses a pretrained iBOT model and learns part tokens to guided the masking process in MAE. In contrast, we treat the sparse tokens as the latent vision representation, and trained them for both low level reconstruction and high-level semantics in a fully self-supervised manner, without requiring annotations or image–text supervision.

As a related concept, sparsity traditionally refers to vectors or matrices with mostly zero entries, and has been extensively studied in signal processing and machine learning. Classical approaches include compressed sensing (Donoho, 2006), sparse coding and dictionary learning (Olshausen & Field, 1997; Mairal et al., 2008; Tošić & Frossard, 2011), and regularization techniques such as the Lasso (Tibshirani, 1996). Sparsity has also been exploited in neural networks through sparse autoencoders (Ng et al., 2011) and structured pruning methods (Anwar et al., 2017). Our notion of sparse representation differs: rather than enforcing zeros in dense feature maps, we learn a compact set of informative tokens, each encoding semantically meaningful features for downstream tasks.

**Low-rank Representation** Low-rank representation is a long-standing idea in machine learning, typically assuming that high-dimensional data lie on a low-dimensional manifold. Classical ap-

proaches include PCA, low-rank matrix recovery (Candes & Plan, 2009), and dictionary learning or sparse coding (Olshausen & Field, 1997; Mairal et al., 2008; Tošić & Frossard, 2011), where signals are expressed through a small set of basis vectors. In deep learning, low-rank constraints have been widely applied for efficiency: for example, low-rank factorization of neural network weights (Sainath et al., 2013), low-rank approximations of attention maps (Katharopoulos et al., 2020), and parameter-efficient fine-tuning methods such as LoRA (Hu et al., 2022). In contrast to these works, STELLAR applies low-rank factorization directly to the feature map from a single image, disentangling spatial and semantic information.

## 5 EXPERIMENTS

We train STELLAR on ImageNet-1K (Deng et al., 2009) in a self-supervised setting. The encoder is a vanilla ViT (Dosovitskiy et al., 2020) augmented with 8–24 learnable latent queries that produce sparse tokens (without positional encoding). The [CLS] token is retained but not used for reconstruction. A lightweight 6-layer ViT serves as the decoder, predicting either MaskGIT-VQGAN tokens (Esser et al., 2021; Chang et al., 2022) or raw pixels (ablation). We initialize the ViT backbone with MAE pre-trained weights to accelerate training, enabling the model to focus on the sparse latent queries, projection layers, and decoder. Alternatively, we experimented with a momentum encoder warm-up. MAE initialization is used by default, with other methods analyzed in our ablations.

### 5.1 SPARSE TOKENS FOR UNDERSTANDING AND RECONSTRUCTION

We evaluate reconstruction with FID (Heusel et al., 2017) and representation quality with linear probing on mean-pooled sparse features. Table 1 and Figure 3 shows results across token counts (8, 16, 24), compared with TiTok (Yu et al., 2024) and MAE (He et al., 2022). STELLAR achieves strong reconstruction even with few tokens (rFID = 3.68 with 8 tokens; 2.60 with ViT-H, 16 tokens), approaching MaskGIT-VQGAN (2.28) without decoder finetuning. For linear probing, STELLAR maintains robust accuracy and does not drop with more tokens as in TiTok (Yu et al., 2024). Reconstruction improves with the number of token, but plateau after 16. We used 16 tokens in all other experiments unless specified. We also see in Figure 3 that STELLAR preserves the location of the objects. Interestingly, it also automatically removes the dark edge in the bottom example, indicating it is reconstructing from high-level semantics rather than memorizing low-level details. Overall, STELLAR balances efficient reconstruction and discriminative understanding at high quality.

Table 1: Reconstruction FID and linear probing accuracy (%) of sparse tokens on IN1K. Model sizes are ViT-B by default, with larger sizes indicated in parentheses.

|  | VQGAN | TiTok | | MAE | | STELLAR (ours) | | | |
|---|---|---|---|---|---|---|---|---|---|
| # tokens | 256 | 32 (L) | 64 | 16 | 32 | 8 | 16 | 24 | 16 (H) |
| rFID ↓ | 2.28 | 2.75 | 1.99 | 150.73 | 131.01 | 3.68 | 3.14 | 3.19 | 2.60 |
| lin. acc. | - | 33.42 | 32.87 | 44.43 | 56.52 | 72.97 | 73.26 | 72.17 | 79.10 |

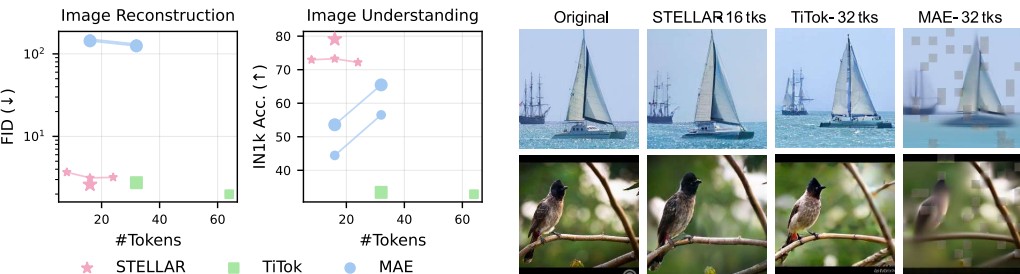

Figure 3: Sparse representation for image understanding and reconstruction. Left: reconstruction quality (FID) and semantic quality (lin. acc.) v.s. number of tokens. Right: reconstruction examples.

Table 2: Segmentation mIOU (%) with linear probing on frozen feature maps. Models are sorted by architecture (*: distilled from larger model), SSL target (gl.: global, de.: dense) and SSL method. **Bold**: best on ImageNet pretraining. Underline: best across same size category.

| **Model** | Arch. | Data | SSL tgt. | Method | ADE20K | CitySc. | VOC |
|---|---|---|---|---|---|---|---|
| *ImageNet trained models* | | | | | | | |
| BYOL | ResNet50 | IN-1K | global | aug. align | 18.43 | 18.66 | 63.89 |
| MoCo v3 | ViT-B | IN-1K | global | contrast. | 29.45 | 25.13 | 74.08 |
| DINO | ViT-B | IN-1K | global | aug. align | 26.87 | 26.82 | 79.29 |
| MSN | ViT-B | IN-1K | global | mask align | 26.66 | 25.39 | 68.59 |
| MSN | ViT-L | IN-1K | global | mask align | 17.94 | 20.06 | 56.14 |
| DenseCL | ResNet50 | IN-1K | dense | contrast. | 23.08 | 18.63 | 70.95 |
| BEIT | ViT-B | IN-22K | dense | MIM token | 11.58 | 18.90 | 27.44 |
| BEIT | ViT-L | IN-22K | dense | MIM token | 12.64 | 20.37 | 25.48 |
| SimMIM | Swin-B | IN-1K | dense | MIM pixel | 12.46 | 17.23 | 35.14 |
| MAE | ViT-B | IN-1K | dense | MIM pixel | 30.91 | 29.44 | 76.43 |
| MAE | ViT-L | IN-1K | dense | MIM pixel | 34.36 | 32.53 | 77.79 |
| MAE | ViT-H | IN-1K | dense | MIM pixel | 36.16 | **35.21** | 78.07 |
| SemMAE | ViT-B | IN-1K | dense | MIM pixel | 3.52 | 25.48 | 48.33 |
| Data2Vec | ViT-B | IN-1K | dense | MIM latent | 22.03 | 23.49 | 61.33 |
| SiameseIM | ViT-B | IN-1K | dense | MIM latent | 29.24 | 26.52 | 81.38 |
| IJEPA | ViT-H | IN-1K | dense | MIM latent | 21.57 | 18.59 | 74.13 |
| iBOT | ViT-B | IN-1K | gl.+de. | align+MIM | 31.78 | 25.69 | 77.06 |
| iBOT | ViT-L | IN-1K | gl.+de. | align+MIM | 33.26 | 26.37 | 77.57 |
| **STELLAR** | ViT-B | IN-1K | sparse | align+recon. | 31.33 | 27.74 | 81.83 |
| **STELLAR** | ViT-L | IN-1K | sparse | align+recon. | 34.02 | 31.32 | **85.90** |
| **STELLAR** | ViT-H | IN-1K | sparse | align+recon. | **36.66** | 33.30 | 85.66 |
| *Larger scale training* | | | | | | | |
| AIM | ViT-600M | DFN-2B+ | dense | autogressive | 29.00 | 27.04 | 64.55 |
| AIM | ViT-1B | DFN-2B+ | dense | autogressive | 29.59 | 27.05 | 63.90 |
| DINOv2 | ViT-B* | LVD142M | gl.+de. | align+MIM | 40.10 | 34.66 | 89.52 |
| DINOv2 | ViT-L* | LVD142M | gl.+de. | align+MIM | 40.45 | 32.07 | 89.19 |

## 5.2 Fine-grained Image Understanding

Since sparse representations are designed to capture complex scenes with multiple objects, we evaluate STELLAR on semantic segmentation via linear probing. We report results on ADE20K (Zhou et al., 2017b), Cityscapes (Cordts et al., 2016), and Pascal VOC (Everingham et al., 2010), comparing against other ImageNet-pretrained SSL models. For broader context, we also include AIM (El-Nouby et al., 2024) and DINOv2 (Oquab et al., 2023), which leverage substantially larger training corpora (100–1000× more images).

As shown in Table 2, STELLAR achieves strong fine-grained understanding despite not applying SSL objectives directly to dense feature maps. Sparse token modeling implicitly organizes the feature map into semantic regions: to reconstruct the image, each token must encode information covering all spatial parts of the scene, resulting in region-aware representations. While STELLAR trails large-scale models such as DINOv2, it surpasses iBOT (Zhou et al., 2021), a closely related SSL method trained on ImageNet-1K.

## 5.3 Global Image Understanding

We assess global image understanding by linear probing on the average-pooled sparse features for classification. Benchmarks include ImageNet-1K (IN1K), Oxford-IIIT Pet (Pets) (Parkhi et al., 2012), Food-101 (Food) (Bossard et al., 2014), and GlaS (Sirinukunwattana et al., 2016) for cancer grade classification in histopathology. Results are summarized in Table 3. For all baseline SSL models, we used mean pooling on the representations where the corresponding SSL method was performed (column "Rep."). We analyzed the effect of token choices in appendix. STELLAR achieves

Table 3: Classification accuracy (%) with linear probing on frozen average pooled representation. Models are sorted by architecture (*: distilled from larger model), SSL space and SSL target (gl.: global, de.: dense). **Bold**: best on ImageNet pretraining. Underline: best across same size category.

| **Model** | Arch. | Data | SSL space | Rep. | IN1K | Pets | Food | GlaS |
|---|---|---|---|---|---|---|---|---|
| *ImageNet trained models* | | | | | | | | |
| BYOL | ResNet50 | IN-1K | latent | global | 70.39 | 82.77 | 64.57 | 95.00 |
| MoCo v3 | ViT-B | IN-1K | latent | global | 74.31 | 91.14 | 77.47 | **97.50** |
| DINO | ViT-B | IN-1K | latent | global | 76.46 | **93.84** | **79.28** | 95.00 |
| MSN | ViT-B | IN-1K | latent | global | 73.65 | 75.91 | 68.93 | 92.50 |
| MSN | ViT-L | IN-1K | latent | global | 59.78 | 2.73 | 24.30 | 53.75 |
| DenseCL | ResNet50 | IN-1K | latent | dense | 61.10 | 72.99 | 59.16 | 85.00 |
| Data2Vec | ViT-B | IN-1K | latent | dense | 54.90 | 26.47 | 34.40 | 73.75 |
| SiameseIM | ViT-B | IN-1K | latent | dense | 74.97 | 91.61 | 71.01 | 91.25 |
| IJEPA | ViT-H | IN-1K | latent | dense | 71.72 | 84.68 | 70.34 | 87.50 |
| iBOT | ViT-B | IN-1K | latent | gl.+de. | 71.58 | 80.16 | 75.92 | 92.50 |
| iBOT | ViT-L | IN-1K | latent | gl.+de. | 74.44 | 82.04 | 77.86 | 96.25 |
| BEIT | ViT-B | IN-22K | image | dense | 32.94 | 36.20 | 54.49 | 90.00 |
| BEIT | ViT-L | IN-22K | image | dense | 36.77 | 36.71 | 56.03 | 90.00 |
| SimMIM | Swin-B | IN-1K | image | dense | 24.77 | 27.39 | 40.94 | 77.50 |
| MAE | ViT-B | IN-1K | image | dense | 66.32 | 81.58 | 70.40 | 93.75 |
| MAE | ViT-L | IN-1K | image | dense | 73.09 | 84.30 | 76.22 | 95.00 |
| MAE | ViT-H | IN-1K | image | dense | 75.22 | 84.96 | 78.36 | 95.00 |
| SemMAE | ViT-B | IN-1K | image | dense | 43.84 | 56.99 | 58.90 | 92.50 |
| TiTok-64 | ViT-B | IN-1K | image | sparse | 32.87 | 42.06 | 43.68 | **97.50** |
| TiTok-32 | ViT-L | IN-1K | image | sparse | 33.42 | 27.83 | 38.83 | 78.75 |
| **STELLAR** | ViT-B | IN-1K | img.+lat. | sparse | 73.26 | 89.70 | 74.09 | 95.00 |
| **STELLAR** | ViT-L | IN-1K | img.+lat. | sparse | 76.94 | 92.53 | 74.78 | **97.50** |
| **STELLAR** | ViT-H | IN-1K | img.+lat. | sparse | **79.10** | 92.53 | 77.43 | 92.50 |
| *Larger scale training* | | | | | | | | |
| AIM | ViT-600M | DFN-2B+ | image | dense | 63.78 | 64.68 | 75.19 | 98.75 |
| AIM | ViT-1B | DFN-2B+ | image | dense | 66.86 | 64.21 | 77.96 | 96.25 |
| DINOv2 | ViT-B* | LVD142M | latent | gl.+de. | 81.14 | 90.92 | 90.35 | 97.50 |
| DINOv2 | ViT-L* | LVD142M | latent | gl.+de. | 82.89 | 91.31 | 92.28 | 98.75 |

the highest accuracy on IN1K at large model scale, but smaller variants underperform methods such as DINO, which explicitly optimize for global representations. Since sparse representations do not model the image as a single holistic concept, averaging token features can dilute discriminative information, which is particularly detrimental on object-centric datasets like Pets and Food.

Interestingly, on histopathology images, which involve complex tissue microenvironments, STELLAR achieves the best performance without task-specific tuning. We also find that TiTok (Yu et al., 2024), another sparse-reconstruction model, performs competitively when using a larger number of tokens, suggesting that sparse image modeling is well suited for domains requiring fine-grained global understanding. Taken together with our segmentation experiments, these results indicate that STELLAR excels at modeling complex, multi-object scenes, while global classification on simple object-centric datasets remains more challenging.

## 5.4 ABLATION ANALYSIS

**Low-rank approximated reconstruction.** As shown in Table 4, removing the low-rank reconstruction objective reduces both global and fine-grained understanding. Since the remaining objectives resemble typical SSL methods, the model still retains reasonable global performance, but fine-grained understanding suffers more. This indicates that low-rank reconstruction encourages sparse tokens to serve as holistic representations covering the entire image.

Table 4: Ablation on IN1K linear probing accuracy (%) and ADE20K linear probing mIOU (%).

| Recon. | Cluster | Set align | CLS align | KoLeo | IN1K lin. | ADE20K lin. |
|--------|---------|-----------|-----------|-------|-----------|-------------|
| *Default* | | | | | | |
| ✓ | ✓ | ✓ | ✓ | ✓ | 73.26 | 31.33 |
| *Ablation model versions* | | | | | | |
| ✗ | ✓ | ✓ | ✓ | ✓ | 72.44 (-0.82) | 29.94 (-1.39) |
| ✓ | ✗ | ✗ | ✗ | ✓ | 52.07 (-21.19) | 20.46 (-10.87) |
| ✓ | ✓ | ✗ | ✗ | ✓ | 2.73 (-70.53) | 1.93 (-29.39) |
| ✓ | ✗ | ✓ | ✓ | ✓ | 42.14 (-31.12) | 18.90 (-12.43) |
| ✓ | ✓ | ✓ | ✗ | ✓ | 70.79 (-2.47) | 30.20 (-1.12) |
| ✓ | ✓ | ✓ | ✓ | ✗ | 72.05 (-1.21) | 30.10 (-1.23) |

**Concept clustering.** Eliminating online clustering and set alignment leads to a sharp drop in understanding, highlighting the necessity of structuring sparse tokens into view-invariant concepts. Notably, low-rank reconstruction with KoLeo regularization alone already surpasses image-reconstruction-based methods such as MAE, SimMIM, BEIT, and TiTok, further demonstrating that the proposed low-rank approximated reconstruction itself induces rich semantics.

**Set alignment.** When training with only reconstruction and clustering, the model collapses, underscoring the critical role of set concepts alignment. Additional alignment on the CLS token primarily benefits global classification but has limited effect on fine-grained understanding. Finally, KoLeo regularization consistently improves both global and fine-grained tasks at similar level. Interestingly, the absent of either concept clustering or set alignment led to a sharp drop in performance.

**Model Initialization** We ablated model initialization and training strategy in Table 5. When initializing the ViT part of the encoder from MAE or DINO while training all other modules from scratch, we observed similar reconstruction and global semantics quality. MAE yields better fine-grained feature map on segmentation task, while both outperformed their initial ViT. When training all modules from scratch, it is necessary to warmup with EMA momentum encoder (150 epochs) before switching to standard training. With a total of 225 epochs, the reconstruction and segmentation quality catch up with using SSL initialized ViT, while the global semantic quality trails behind.

Table 5: Ablation on model initialization and training strategy

| | MAE | DINO | MAE init. | DINO init. | EMA only | EMA+75ep. |
|--------------------|-------|-------|-----------|------------|----------|-----------|
| Recon. FID (↓) | - | - | 3.14 | 3.31 | 3.69 | 3.21 |
| Class. Acc. | 66.32 | 76.46 | 73.26 | 73.31 | 58.83 | 65.28 |
| Seg. mIOU | 30.91 | 26.87 | 31.33 | 28.17 | 26.79 | 28.10 |

## 6 DISCUSSION AND CONCLUSION

We demonstrate that sparse visual representation, when structured in form of low-rank factorization, can simultaneously support high-fidelity reconstruction and rich semantic understanding. By disentangling *what* and *where*, STELLAR learns compact tokens that transfer effectively to both global and fine-grained tasks, surpassing prior methods. The sparse image modeling framework effectively shapes the feature map into semantic regions, without direct loss on the dense representation.

Nevertheless, several limitations remain. First, our experiments are conducted primarily on ImageNet-1K scale; extending to larger pretraining corpora is likely to further narrowing the gap with large-scale models such as DINOv2. Second, the current framework adopts a minimalist architecture. Exploring more expressive decoders or hybrid architectures could enhance reconstruction fidelity. Finally, while STELLAR naturally provides a compact interface for multimodal alignment, we only experimented simple alignment in appendix, leaving systematic integration with language models to future work. Overall, our results highlight sparse tokens as a promising direction for unifying efficiency, interpretability, and semantic richness in visual representation learning.

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

## A  STATEMENTS

### A.1  ETHICS STATEMENT

This work adheres to the ICLR Code of Ethics. Our research is based on publicly available datasets (e.g., ImageNet-1K, ADE20K, Cityscapes, Pascal VOC) and does not involve human subjects, private data, or personally identifiable information. We follow standard licensing terms for all datasets used. The proposed framework, STELLAR, is a general-purpose method for self-supervised representation learning and is not designed for harmful or sensitive applications.

### A.2  REPRODUCIBILITY STATEMENT

We have made significant efforts to ensure the reproducibility of our results. Detailed descriptions of the STELLAR framework, training objectives, and experimental setups are provided in the main text and appendix. We report all datasets used, including preprocessing steps and evaluation protocols, and we describe ablation studies to clarify the contribution of each component. We will release the source code and pretrained model checkpoints after the internal review process to further facilitate reproducibility.

### A.3  USE OF LARGE LANGUAGE MODELS

Large language models (ChatGPT) were used exclusively for language refinement, including polishing grammar, phrasing, and clarity of the manuscript. They were not used for research ideation, methodological design, experimental implementation, data analysis, or drawing conclusions. All scientific contributions of this work are entirely by the authors.

## B  IMPLEMENTATION DETAILS

### B.1  STELLAR TRAINING

We trained STELLAR with ViT models at size base, large, and huge, along with the latent queries, projection layers, clustering head, and a 6-layer ViT decoder. In the default setting, we initialized the ViT part in the encoder from public MAE checkpoint, and trained for 150 epochs for STELLAR-B, 100 epochs for STELLAR-L, and 50 epochs for STELLAR-H. We used 16 NVIDIA A100-80GB with batch size 128 each, totaling 2048. We used AdamW(Loshchilov, 2017) with base learning rate $1.5 \times 10^{-4}$ for STELLAR-B, and $5 \times 10^{-5}$ for STELLAR-L and STELLAR-H.

For concept clustering, we used 16384 prototypes for sparse and CLS tokens each. The projector is a 2-layer MLP before the prototype layer. We used 3 steps of Sinkhon-Knopp algorithm. The temperature in sparse-dense cosine similarity softmax is 0.06. We used 6-8 random masked views to align the sparse tokens, and additional 6-8 local crops to align the CLS token.

In the ablation study of model initialization and training strategy (Table 5), we trained the model from scratch and used exponential moving average (EMA) updated momentum encoder to encode the target prototype assignments in the warm-up stage. We EMA updated the full encoder (ViT, latent queries, projection, clustering head with momentum 0.996. The momentum encoder was used to encode a global view of the image into target prototype assignments, for both clustering loss and alignment loss. The masking ratio was 0.6 in the warm-up stage, and 0.8 during standard training. We trained the model with 150 epochs of EMA warm-up and 75 epochs of standard training.

### B.2  EVALUATION PROTOCOL

For STELLAR and all baseline models, we evaluated the frozen feature from the pretrained model with linear probing. We used layer norm in classification tasks, and batch norm in segmentation tasks, followed by a single linear layer predicting the class of the image or patch. For all benchmarks, we split 10% from the training set for validation. We tuned hyper-parameter with learning rate $1 \times 10^{-5}, 2 \times 10^{-5}, 5 \times 10^{-5}, 1 \times 10^{-4}, 2 \times 10^{-4}, 5 \times 10^{-4}, 1 \times 10^{-3}, 2 \times 10^{-3}, 5 \times 10^{-3}, 1 \times 10^{-2}$, and batch size 64, 128, 256, 512, 1024, 2048, 4096, 8192.

As the SSL methods varies across different baseline models, for classification tasks we used the mean-pooled feature from the representations where the corresponding SSL method was performed, e.g. the global CLS token for DINO, and dense patch tokens for MAE. We noticed the linear probing accuracy can vary depending on the pooling choice, and conducted experiments by using different types of tokens for each model, with results in Table 6. We observed that the SSL-ed are typically the best choice for linear probing, except for iBOT, which highly relies on the global CLS token for classification, even though the model was trained with MIM. In contrast, STELLAR and MAE are relatively more robust to token choices.

Table 6: ImageNet-1K linear probing accuracy (%) by pooling different tokens. We mark in **bold** the tokens on which the specific SSL method was applied, and the top accuracy for each method.

| | DINO | | MAE | | iBOT | | | STELLAR (ours) | |
|---|---|---|---|---|---|---|---|---|---|
| tokens | **global** | dense | global | **dense** | **global** | dense | **gl.+de.** | **sparse** | dense |
| lin. acc. | **76.46** | 70.31 | 65.61 | **66.32** | **76.40** | 71.44 | 71.58 | **73.26** | 72.21 |

Table 7: List of baseline models and SSL method type.

| **Model** | Reference | Method | SSL space | SSL tokens |
|---|---|---|---|---|
| BYOL | Grill et al. (2020) | augmentation alignment | latent | global |
| MoCo v3 | Chen et al. (2021) | contrastive learning | latent | global |
| DINO | Caron et al. (2021) | augmentation alignment | latent | global |
| MSN | Assran et al. (2022) | masked alignment | latent | global |
| DenseCL | Wang et al. (2021) | contrastive learning | latent | dense |
| Data2Vec | Baevski et al. (2022) | latent MIM | latent | dense |
| SiameseIM | Tao et al. (2023) | latent MIM | latent | dense |
| IJEPA | Assran et al. (2023) | latent MIM | latent | dense |
| iBOT | Zhou et al. (2021) | align + latent MIM | latent | global+dense |
| BEIT | Bao et al. (2021) | token MIM | image | dense |
| SimMIM | Xie et al. (2022) | pixel MIM | image | dense |
| MAE | He et al. (2022) | pixel MIM | image | dense |
| SemMAE | Li et al. (2022) | pixel MIM | image | dense |
| TiTok | Yu et al. (2024) | reconstruction + clustering | image | sparse |
| AIM | El-Nouby et al. (2024) | autoregressive | image | dense |
| DINOv2 | Oquab et al. (2023) | align + latent MIM | latent | global+dense |

## C ADDITIONAL RESULTS

### C.1 EFFECT OF PRETRAINING DATA

We pretrained separate STELLAR versions on ImageNet-1K, Places365 (Zhou et al., 2017a) and compared their linear probing performance in Table 8.

Table 8: Effect of pretraining data.

| | linear probing acc. | |
|---|---|---|
| Pretraining data | ImageNet-1K | Places 365 |
| ImageNet-1K | 76.94 | 49.25 |
| Places365 | 66.08 | 51.98 |

## C.2 SEMANTICS FROM DIFFERENT FEATURES

We conducted linear probing of different mean-pooled features of different types, and compared in Table 9. Sparse feature showed strongest global understanding quality.

Table 9: Semantics in different features

| Feature | sparse | cls | dense |
|---|---|---|---|
| IN-1K lin. acc (%) | 73.26 | 72.23 | 72.21 |

## C.3 CONCEPT ALIGNMENT WITH LANGUAGE

Inspired by Zhang et al. (2025), we used frozen feature from STELLAR and aligned with the text tower of CLIP (Radford et al., 2021) with a single attention pooled probing layer. The evaluation on vision language tasks with comparison to baseline models are shown in Table 10.

Table 10: Language alignment evaluation.

| | IN-1K 0-shot | | MS COCO | | Winoground | | MMVP |
|---|---|---|---|---|---|---|---|
| | @1 | @5 | T2I | I2T | Text | Image | Avg. |
| MAE | 23.18 | 50.43 | 11.28 | 13.46 | 20.75 | 9.00 | 19.26 |
| iBOT | 50.01 | 80.43 | 20.79 | 29.38 | 24.75 | 12.00 | 18.52 |
| STELLAR | 51.53 | 80.04 | 17.94 | 22.34 | 26.25 | 8.25 | 19.26 |
| CLIP | 72.7 | - | 43.0 | 59.7 | 30.5 | 11.5 | 20.0 |

## C.4 FINETUNING

We performed finetuning for STELLAR on ImageNet-1K classification and ADE20K segmentation, and compared with baseline models. We used the same evaluation protocol as in Sec. B.2, with the backbone unfrozen and finetuned for 75 epochs. We used ViT-B for all models. The finetuning results are shown in Table 12. STELLAR showed consistent performance gain across different tasks, and close to the top model iBOT with slight difference.

Table 11: Finetuning performance in ImageNet-1K classification accuracy and ADE20K segmentation mIOU (%). We show in parentheses the gain over the respective linear probing results.

| Model | ImageNet-1K Acc. | ADE20K mIOU |
|---|---|---|
| DINO | 79.58 (+3.12) | 39.22 (+12.35) |
| MAE | 77.75 (+11.43) | 40.33 (+9.42) |
| iBOT | 80.72 (+9.14) | 42.76 (+10.97) |
| STELLAR | 80.05 (+6.78) | 41.98 (+10.65) |

## C.5 EFFICIENCY ANALYSIS

To analyze the efficiency of the STELLAR framework, we printed the processing time of the main components in the STELLAR framework with one A100 GPU at different batch sizes. Encoding the main global view of the image takes up most of the processing time, followed by encoding the masked views (8 views at 80% masking ratio) and decoding to the original image. The Sinkhorn-Knopp algorithm used for clustering and the Sinkhorn algorithm used in optimal transport matching take up much less amount of time, and their total processing time stay at similar level when increasing the batch size.

In comparison to the Sinkhorn matching algorithm we used in our experiments, we show the processing time using an alternative Hungarian matching algorithm commonly used in previous literature such as Sparse R-CNN (Sun et al., 2021), DETR (Carion et al., 2020) and MaskFormer (Cheng et al., 2021). As the implementation of the exact matching is not scalable with GPU parallelization, it's computational time increases linearly with the batch size. At batch size 64, it is already 6 times of the encoder processing, while the Sinkhorn algorithm is over 100 times faster. For this reason, we added a small entropy regularization term in the bipartite matching objective, allowing us to use the Sinkhorn algorithm for efficient matching with GPU parallelization.

Table 12: Processing time (s) of the main components in the STELLAR framework with one A100 GPU at different batch sizes. In comparison to the Sinkhorn matching algorithm we used in our experiments, we show the processing time using an alternative Hungarian matching algorithm commonly used in previous literature (shown in gray).

| Batch size | 4 | 8 | 16 | 32 | 64 |
|---|---|---|---|---|---|
| Encoder | $8.2 \times 10^{-3}$ | $9.1 \times 10^{-3}$ | $1.4 \times 10^{-2}$ | $2.0 \times 10^{-2}$ | $3.2 \times 10^{-2}$ |
| Decoder | $4.6 \times 10^{-3}$ | $6.8 \times 10^{-3}$ | $8.8 \times 10^{-3}$ | $1.2 \times 10^{-2}$ | $1.5 \times 10^{-2}$ |
| Mask encoding | $7.9 \times 10^{-3}$ | $8.9 \times 10^{-3}$ | $1.1 \times 10^{-2}$ | $1.8 \times 10^{-2}$ | $1.7 \times 10^{-2}$ |
| SK clustering | $3.4 \times 10^{-4}$ | $3.4 \times 10^{-4}$ | $3.4 \times 10^{-4}$ | $3.7 \times 10^{-4}$ | $3.9 \times 10^{-4}$ |
| Sinkhorn matching | $1.4 \times 10^{-3}$ | $1.4 \times 10^{-3}$ | $1.4 \times 10^{-3}$ | $1.4 \times 10^{-3}$ | $1.2 \times 10^{-3}$ |
| Hungarian matching | $5.7 \times 10^{-3}$ | $1.7 \times 10^{-2}$ | $4.0 \times 10^{-2}$ | $9.0 \times 10^{-2}$ | $1.8 \times 10^{-1}$ |