# OpenReview forum: "Self-supervised Sparse Vision Concepts for Image Understanding and Reconstruction"
_ICLR.cc/2026/Conference — Submitted to ICLR 2026_

### Official Review · Reviewer_nvJG · 2025-10-25

**Soundness:** 3
**Presentation:** 3
**Contribution:** 3
**Rating:** 6
**Confidence:** 5

**Summary:**

This paper proposes a self-supervised visual representation learning framework called STELLAR, which uses low-rank matrix factorization to compress images into a small number of sparse tokens. This approach supports both high-quality image understanding and pixel-by-pixel reconstruction without manual annotation. This approach learns representations by decoupling the semantics of "what" and the spatial location of "where." It then uses self-supervised clustering and optimal transport to align concepts from different perspectives. This approach achieves superior reconstruction quality and semantic expressiveness with only eight sparse tokens, surpassing existing sparse modeling methods on multiple downstream tasks.

**Strengths:**

1. The paper is well-motivated, that low-rank matrix factorization is used to generate small but informative sparse visual tokens, decoupling the semantic concepts and spatial localization.

2. The method provides a relatively complete theoretical basis.

3. The experiments are conducted across various tasks, including segmentation, classification, and linear probing, showing great generalization.

**Weaknesses:**

1. In line 728, the paper claim that the model is initialized from MAE checkpoint. However, the MAE pre-trained model already has strong generalization properties and cannot demonstrate the effectiveness of the proposed STELLAR framework. Especially for self-supervised learning, it is particularly difficult to initialize the model from scratch. It is strongly recommended that authors provide pre-training results from scratch.

2. Several self-supervised methods need to be discussed, especially for efficiency: MoCo v3[1], SiameseIM [2], and OCL[3].

3. I wonder the number of learnable latent queries is fixed (8-24) or self-adapted? It is suggested to add ablation experiments about learnable latent queries.

[1] Chen, Xinlei, Saining Xie, and Kaiming He. "An empirical study of training self-supervised vision transformers." ICCV. 2021.

[2] Tao, Chenxin, et al. "Siamese image modeling for self-supervised vision representation learning." CVPR. 2023.

[3] Yang, Xiaoyu, et al. "One Leaf Reveals the Season: Occlusion-Based Contrastive Learning with Semantic-Aware Views for Efficient Visual Representation." ICML 2025.

**Questions:**

See Weakness

---

> ### Author Response · Authors · 2025-11-22
> **Authors’ Response to Reviewer nvJG**
>
> We thank the reviewer for the positive and encouraging assessment of our work. We appreciate the recognition that the paper is well-motivated, that the proposed low-rank formulation provides a coherent and principled way to generate informative sparse visual tokens with theoretical grounding. We are also grateful for the reviewer’s acknowledgement of the breadth of our experimental evaluation, demonstrating strong reconstruction quality and semantic expressiveness. We have carefully revised the manuscript in response to all comments, with changes highlighted in blue, and provide detailed point-by-point responses below.
>
> **Q1**: "In line 728, the paper claim that the model is initialized from MAE checkpoint. However, the MAE pre-trained model already has strong generalization properties and cannot demonstrate the effectiveness of the proposed STELLAR framework. Especially for self-supervised learning, it is particularly difficult to initialize the model from scratch. It is strongly recommended that authors provide pre-training results from scratch."
> - Thank you for this important suggestion. In the revised manuscript (with edits highlighted in blue), we now clearly describe the full training procedure and provide an ablation of model initialization and training strategies in Table 5.
> - **Initialization and training strategy**
>   - We initialize the ViT backbone with MAE pre-trained weights to accelerate convergence and allow the training to focus on learning the sparse latent queries, projection layers, and decoder.
>   - As an alternative, we also experimented with a short warm-up stage using a momentum encoder to generate target assignments before switching to the standard training schedule. We analyzed outcome of different initialization and training strategies in our ablations.
> - **Effect of different initializations**
>   - When initializing only the ViT backbone from MAE or DINO and training all other STELLAR-specific modules from scratch, we observe similar reconstruction quality and global semantic performance. MAE initialization provides somewhat better fine-grained feature maps for segmentation tasks, although both methods outperform the vanilla ViT from which they start.
>   - When training all modules from scratch, it is necessary to include a momentum-encoder warm-up (150 epochs) before switching to standard training (75 epochs). We observed that with a total of 225 epochs, reconstruction and segmentation quality catch up to the SSL-initialized backbone, while global semantic quality remains lower.
> - **Clarifying how STELLAR relates to MAE** While we observed that it is helpful to use an SSL initialized ViT for the model to learn the desired sparse features for unified representation, it does not conflict the proposed method:
>   - STELLAR produces sparse representations for both reconstruction and semantics, whereas MAE and other ViT models produces dense patch embeddings.
>   - MAE initialization uses no additional data or supervision beyond ImageNet-1K and does not introduce new information into STELLAR.
>   - The choice of MAE is purely practical—MAE (and DINO) are widely used and publicly available for ViT initialization, enabling stable training and reproducibility.
>   - The key contribution of STELLAR lies in its semantic-spatial factorization form in the latent space, and its ability to learn sparse tokens that jointly support pixel-level reconstruction and high-level semantic understanding in a fully self-supervised manner.
>   - We have clarified these points in the revised manuscript to avoid confusion and to show that warm-starting the ViT backbone is a practical choice to achieve the goal, not a conceptual requirement for STELLAR.
>
> **Q2**: Several self-supervised methods need to be discussed, especially for efficiency: MoCo v3[1], SiameseIM [2], and OCL[3].
> - We have added comparisons with MoCo v3 and SiameseIM. (Table 2, 3) OCL is not open-sourced accessible yet for comparison but we included it in the Related Work section.
>
> **Q3**: "I wonder the number of learnable latent queries is fixed (8-24) or self-adapted? It is suggested to add ablation experiments about learnable latent queries."
> - The number of the learnable latent queries is a hyperparameter. In the revised manuscript, we improved the presentation of this ablation with a an additional table (Table 1) and examines the effect of different number of learnable latent queries:
>   - _For linear probing, STELLAR maintains robust accuracy and does not drop with more tokens as in TiTok (Yu et al., 2024). Reconstruction improves with the number of token, but plateau after 16. We used 16 tokens in all other experiments unless specified._
>
> We hope that the additional experiments and clarifications improve the clarity and strengthen the contribution of our work. If the revisions satisfactorily address the reviewer’s concerns, we would be grateful for a corresponding update of the review score. Thank you again for your thoughtful feedback!

---

### Official Review · Reviewer_ypJX · 2025-10-30

**Soundness:** 3
**Presentation:** 3
**Contribution:** 2
**Rating:** 4
**Confidence:** 4

**Summary:**

The authors propose a self-supervised framework for sparse visual representations. Specifically, the authors represent images with a small number of sparse tokens via low-rank matrix factorization that disentangles semantic and spatial information. Several additional losses, including clustering loss and set alignment loss, are also introduced. Experimental results on image reconstruction and understanding tasks demonstrate the effectiveness of the proposed method.

**Strengths:**

-	The proposed method is well motivated. Sparse tokens are indeed a promising way for unifying efficiency, interpretability, and semantic richness in visual representations.
-	The paper is generally well-written and easy to follow.
-	The experiments are extensive and the results seem promising, while some parts need to be improved (see weaknesses).

**Weaknesses:**

-	Some results are missing exact descriptions. For example, Figure 3 is a bit confusing for me. What do the three colors represent? Besides, what specific number of tokens are used in Table 1 and Table 2? It seems that the authors do not ablate the effect of the number of tokens for understanding tasks.
-	According to Table 3, it seems that adding clustering decreases the performance significantly (row 2 vs. row 3 in `Ablation model versions`). Could the authors provide a justification for this?
-	The proposed method uses optimal transport matching and Sinkhorn-Knopp algorithms, which could be potentially computationally expensive. It would be better to provide a computational cost comparison (e.g., training time, memory cost) with previous sparse representation learning methods.
-	Apart from linear probing that serves as a feature extractor, it would be better to provide full fine-tuning results as well, considering fine-tuning is also one of the important benchmarks to evaluate whether the learned representations can serve as a good initialization for downstream tasks.

**Questions:**

I am concerned about the questions mentioned above. Given the current status of the paper, I am leaning towards borderline reject and hope the authors could address my concerns during the rebuttal.

---

> ### Author Response · Authors · 2025-11-22
> **Authors’ Response to Reviewer ypJX**
>
> We thank the reviewer for highlighting our motivation, method design, and experimental results. In particular, we appreciate the recognition that sparse vision representation as a promising direction to unify efficiency, interpretability, and semantic richness in visual representations, supported by extensive experiments. We also thank the reviewer for the thoughtful suggestions regarding clarity and completeness. We have carefully revised the manuscript to address all the points, with changes highlighted in blue for better visibility. Our detailed point-by-point responses are provided below.
>
> Q1: "Some results are missing exact descriptions. For example, Figure 3 is a bit confusing for me. What do the three colors represent? Besides, what specific number of tokens are used in Table 1 and Table 2? It seems that the authors do not ablate the effect of the number of tokens for understanding tasks."
> - Thank you for raising this point. We have improved Figure 3 with clearer formatting and visualization to better present the experimental findings. To provide a more complete presentation of the results, we also added an additional Table (now Table 1) that reports detailed quantitative comparisons, including an ablation over the number of sparse tokens.
>
> | # tokens          | VQGAN (256) | TiTok-L (32) | TiTok (64) | MAE (16) | MAE (32) | STELLAR (8) | STELLAR (16) | STELLAR (24) | STELLAR-H (16) |
> | ----------------- | -------------- | --------------- | ----------- | --------- | --------- | ------------ | ------------- | ------------- | ----------------- |
> | **rFID ↓**        | 2.28           | 2.75            | 1.99        | 150.73    | 131.01    | 3.68         | 3.14          | 3.19          | 2.60              |
> | **lin. acc. (%)** | –              | 33.42           | 32.87       | 44.43     | 56.52     | 72.97        | 73.26         | 72.17         | 79.10             |
>
> - For clarity, we now explicitly state in Section 5.1 that: "We use 16 sparse tokens for all experiments unless otherwise specified."
>
> Q2: " According to Table 3, it seems that adding clustering decreases the performance significantly (row 2 vs. row 3 in Ablation model versions). Could the authors provide a justification for this?"
> - Thank you for the insightful question. To clarify the role of the clustering loss, we added an additional ablation where only the clustering loss term was removed, while all other components of the objective are kept unchanged. The new results (now included in Table 4) show that removing clustering loss leads to a sharp drop in performance, confirming that clustering contributes positively when combined with the other loss terms.
> - This observation together with all previous ablation experiments shows an interesting interaction in the training dynamics. The clustering loss and the alignment loss need to be present at the same time in order to learn rich semantics, but not in separate. In particular, the low performance with clustering+reconstruction indicates a form of feature collapse, while the other low performing cases 1. reconstruction only and 2. reconstruction+alignment indicates non-collapsing suboptimal solution. Without the augmentation from the set alignment loss, clustering along can lead the model to find trivial solution (collapse), but with the alignment loss it helps the model to build more structures concept space.
>
> | Recon. | Cluster | Set align | CLS align | KoLeo | IN1K lin. | ADE20K lin. |
> |--------|---------|-----------|-----------|--------|------------|--------------|
> | *Default* |||||||
> | **✔︎** | **✔︎** | **✔︎** | **✔︎** | **✔︎** | **73.26** | **31.33** |
> | *Ablation model versions* |||||||
> | **✘** | **✔︎** | **✔︎** | **✔︎** | **✔︎** | 72.44 (−0.82) | 29.94 (−1.39) |
> | **✔︎** | **✘** | **✘** | **✘** | **✔︎** | 52.07 (−21.19) | 20.46 (−10.87) |
> | **✔︎** | **✔︎** | **✘** | **✘** | **✔︎** | 2.73 (−70.53) | 1.93 (−29.39) |
> | **✔︎** | **✘ (new)** | **✔︎** | **✔︎** | **✔︎** | 42.14 (−31.12) | 18.90 (−12.43) |
> | **✔︎** | **✔︎** | **✔︎** | **✘** | **✔︎** | 70.79 (−2.47) | 30.20 (−1.12) |
> | **✔︎** | **✔︎** | **✔︎** | **✔︎** | **✘** | 72.05 (−1.21) | 30.10 (−1.23) |

---

> ### Author Response · Authors · 2025-11-22
> **Authors’ Response to Reviewer ypJX (cont'd)**
>
> Q3: "The proposed method uses optimal transport matching and Sinkhorn-Knopp algorithms, which could be potentially computationally expensive. It would be better to provide a computational cost comparison (e.g., training time, memory cost) with previous sparse representation learning methods."
> - Thanks for raising the question. Efficiency was indeed an important consideration in designing our framework. We’ve now included the detailed computation evaluation in Appendix C.5, under different batch sizes. We added the following discussion about the comparison table:
>   - _In particular, the Sinkhorn-Knopp algorithm used for clustering and the Sinkhorn algorithm used in optimal transport matching take up much less amount of time compared to the encoder and decoder part, and their total processing time stay at similar level when increasing the batch size. Therefore, the extra computational burden is low._
>   - _In comparison to the Sinkhorn matching algorithm we used in our experiments, we show the processing time using an alternative Hungarian matching algorithm commonly used in previous literature such as Sparse R-CNN, DETR and MaskFormer. As the implementation of the exact matching is not scalable with GPU parallelization, it's computational time increases linearly with the batch size. At batch size 64, it is already 6 times of the encoder processing, while the Sinkhorn algorithm is over 100 times faster. For this reason, we added a small entropy regularization term in the bipartite matching objective, allowing us to use the Sinkhorn algorithm for efficient matching with GPU parallelization._
>
> | Batch size                  | 4          | 8          | 16         | 32         | 64         |
> | --------------------------- | ---------- | ---------- | ---------- | ---------- | ---------- |
> | **Encoder**                 | 8.2×10⁻³   | 9.1×10⁻³   | 1.4×10⁻²   | 2.0×10⁻²   | 3.2×10⁻²   |
> | **Decoder**                 | 4.6×10⁻³   | 6.8×10⁻³   | 8.8×10⁻³   | 1.2×10⁻²   | 1.5×10⁻²   |
> | **Mask encoding**           | 7.9×10⁻³   | 8.9×10⁻³   | 1.1×10⁻²   | 1.8×10⁻²   | 1.7×10⁻²   |
> | **SK clustering**           | 3.4×10⁻⁴   | 3.4×10⁻⁴   | 3.4×10⁻⁴   | 3.7×10⁻⁴   | 3.9×10⁻⁴   |
> | **Sinkhorn matching**       | 1.4×10⁻³   | 1.4×10⁻³   | 1.4×10⁻³   | 1.4×10⁻³   | 1.2×10⁻³   |
> | *Hungarian matching (comparison)* | *5.7×10⁻³* | *1.7×10⁻²* | *4.0×10⁻²* | *9.0×10⁻²* | *1.8×10⁻¹* |
>
> Q4: "Apart from linear probing that serves as a feature extractor, it would be better to provide full fine-tuning results as well, considering fine-tuning is also one of the important benchmarks to evaluate whether the learned representations can serve as a good initialization for downstream tasks."
> - Thank you for this important suggestion. We have now conducted full finetuning experiments for both classification and segmentation, with results and complete implementation details in Appendix C.4. Across the downstream tasks, STELLAR achieves best or near-best performance compared, confirming that the model also provides a strong initialization for end-to-end adaptation.
> - As the primary goal of STELLAR is to learn strong frozen representations that support both reconstruction and semantic understanding, our main experiments emphasize the performance of the self-supervised features directly. For completeness, we provide the finetuning results in the supplementary material, and we believe these additional experiments further strengthen the applicability of our method.
>
> | Model              | ImageNet-1K Acc.  | ADE20K mIoU        |
> | ------------------ | ----------------- | ------------------ |
> | **DINO**           | 79.58 (+3.12)     | 39.22 (+12.35)     |
> | **MAE**            | 77.75 (+11.43)    | 40.33 (+9.42)       |
> | **iBOT**           | 80.72 (+9.14)     | 42.76 (+10.97)     |
> | **STELLAR (ours)** | 80.05 (+6.78) | 41.98 (+10.65) |
>
>
> We thank the reviewer again for the detailed and constructive feedback. With the revision based on all suggestions and questions, we believe the updated manuscript substantially improves the clarity, completeness, and presentation of our contributions. We hope these revisions address the concerns raised and that the improvements are reflected in the final assessment. Thank you for your time and thoughtful review!

---

### Official Review · Reviewer_A18a · 2025-11-01

**Soundness:** 2
**Presentation:** 2
**Contribution:** 1
**Rating:** 2
**Confidence:** 5

**Summary:**

This paper propose a self-supervised learning recipe in the vision domain. The authors introduce a self-supervised framework named STELLAR that replaces dense tokens with a small set of sparse tokens. STELLAR  Specifically, STELLAR learns a tiny set of sparse concept tokens and per-patch localization weights, then approximate a feature of the given input data using them, letting the model reconstruct images and transfer semantics with just a few tokens The authors employ transport (e.g., sinkhorn) to cluster the sparse visual concepts from the dataset into prototypes. Experimental results including reconstruction, linear probing, segmentation validates the effectiveness of the proposed method.

**Strengths:**

* Introduce an idea of learning sparse visual concept
* Provided an ablation study on each component

**Weaknesses:**

* Concern on technical novelty
    * Isn't it highly dependent on the MAE initialization?

* Concern on its motivation and intuition
    * From my perspective, the motivation of the proposed method seems similar to that of SemMAE [1] since SemMAE also tried to learn the find-grained information of the semantics (e.g., information of the objects' part) of the. Could the authors clarify the difference between STELLAR and SemMAE in terms of the motivation and intuition?

* The comparison in Table 2 is not reliable. Some baselines are reported far below than the results in their original papers
    * e.g., according to Table 1 in the iBOT paper, linear probing accuracies for DINO and iBOT exceed 80.0%, surpassing all linear probing results in the Table 1 in the authors' paper
    * Note that the epochs reported in the iBOT paper are effective epochs that account for multi-crop, not the actual pre-training epochs used by iBOT or DINO. They actually pre-trained only for 300 or 400 epochs.
    * Moreover, the proposed method employ MAE model parameters for initialization, which is not fair with other baselines. Also, the total epoch should be regarded as 1600 (MAE pre-training epochs) + 150/100/50 (STELLA post-training epochs) = 1750 / 1700 / 1650.

* A lot of recent self-supervised learning methods are missing.
    * e.g., the references below [1-22]
    * The proposed method should also be compared with these methods

* Some core evaluation tasks are missing
    * e.g., Detection and segmentation on COCO

* Important comparison results are missing
    * Fine-tuning performance comparison is very important in self-supervised learning area on visual data. However,
        * I'm also suspecting that the proposed design may improve only linear-probing performance rather than fine-tuning performance. This concern is amplified by the utilization of the MAE initialization since MAE is well-known to show strong fine-tuning performance and weak linear-probing performance.



[1] Li et al., SemMAE, NeurIPS 2022

[2] Mishra et al., CAN, arXiv 2022

[3] Baevski et al., data2vec, ICML 2022

[4] Chen et al., SdAE, ECCV 2022

[5] Assran et al., MSN, ECCV 2022

[6] Dong et al., BootMAE, ECCV 2022

[7] Baevski et al., data2vec2.0, ICML 2023

[8] Wang et al., AdPE, arXiv 2023

[9] Wu et al., ExtreMa, TMLR 2023

[10] Huang et al., CMAE, TPAMI 2023

[11] Yi et al., ConMIM, ICLR 2023

[12] Yi et al., RC-MAE, ICLR 2023

[13] Chen et al., MixedAE, CVPR 2023

[14] Tao et al., SIM, CVPR 2023

[15] Wang et al., HPM, CVPR 2023

[16] Huang et al., MIRL, NeurIPS 2023

[17] Fu et al., CrossMAE, arXiv 2024

[18] Kim et al., LUT, ECCV 2024

[19] Liu et al., dBOT, ICLR 2024

**Questions:**

What happens if STELLAR does not use the MAE initialization?

---

> ### Author Response · Authors · 2025-11-23
> **Authors’ Response to Reviewer A18a - 1**
>
> We thank the reviewer for the thoughtful comments and for highlighting the value of learning sparse visual concepts and the usefulness validated by our ablation studies. We appreciate the opportunity to clarify the goals and contributions of STELLAR, and we have incorporated additional experimental results and revisions (highlighted in blue) to strengthen the presentation and rigor of the work. Our point-by-point responses addressing all concerns are provided below.
> 1. "Concern on technical novelty - Isn't it highly dependent on the MAE initialization?"
> - Thank you for raising this concern. In the revised manuscript (with edits highlighted in blue), we have clarified the full procedure and added an ablation of initialization and training strategies (Table 5).
>   - **Initialization and training strategy** We initialize the ViT backbone with publicly available MAE weights purely as a practical choice, allowing the model to focus on learning the sparse latent queries, spatial projection layers, decoder, and SSL objectives. As an alternative, we also evaluate a short warm-up stage using a momentum encoder before switching to standard training. Both strategies are now described and compared in the manuscript.
>   - **Effects of different initializations** Our ablation results show that STELLAR is not dependent on MAE initialization:
>     - MAE vs. DINO initialization: When initializing only the ViT backbone from either MAE or DINO and training all STELLAR-specific modules from scratch, we observe similar reconstruction quality and global semantic performance. MAE provides slightly better fine-grained feature maps for segmentation, but both outperform the results from the original ViT.
>     - Training all modules from scratch: When no prior SSL initialization is used, STELLAR still learns meaningful sparse tokens. A momentum-encoder warm-up (150 epochs) followed by standard training (75 epochs) is necessary for stability. With a total of 225 epochs, reconstruction and segmentation performance catch up to SSL-initialized backbones, although global semantic quality remains lower.
>     - These results demonstrate that MAE initialization provides efficiency but is not a conceptual requirement for STELLAR.
>   - We emphasize that using MAE as the ViT initialization doesn't conflict the claim in that:
>     - MAE produces dense patch embeddings, whereas STELLAR produces a sparse semantic–spatial factorization, which is structurally different.
>     - MAE initialization uses no additional data or labels beyond ImageNet-1K and introduces no new supervision into STELLAR.
>     - The contribution of STELLAR lies in the representation form: learning sparse concept tokens and localization weights that jointly support pixel-level reconstruction and semantic understanding. Warm-starting the ViT backbone is therefore a practical engineering choice, consistent with many state-of-the-art SSL methods that reuse pretrained components (e.g., BEiT [2] using DALL-E as image reconstruction target, SemMAE [1] using iBOT to guide the masking process, BEiT v2 [3] using DINO/CLIP for semantic reconstruction), and does not diminish the novelty of the representation learning framework.
> - These clarifications and ablations have been added to the revised manuscript to avoid any misinterpretation regarding dependence on MAE initialization.
>
> 2. "Concern on its motivation and intuition - From my perspective, the motivation of the proposed method seems similar to that of SemMAE [1] since SemMAE also tried to learn the find-grained information of the semantics (e.g., information of the objects' part) of the. Could the authors clarify the difference between STELLAR and SemMAE in terms of the motivation and intuition?"
> - Thank you for raising this point. We agree that clarifying the relationship to SemMAE is important, and we have now included a comparison in the Related Work section, and also added SemMAE as an additional baseline in our experiments.
> - SemMAE shares the intuition that images contain fine-grained semantic parts that can be captured with a small set of learnable tokens. However, the motivation and the role of these tokens differ fundamentally between SemMAE and STELLAR:
>   - SemMAE uses part tokens to guide the masking strategy of MAE. These tokens are auxiliary: they modulate which patches MAE should mask, but they are not used as the learned representation of the image. Its training objective is still tied to MAE’s reconstruction of dense patch embeddings.
>   - In contrast, STELLAR treats its sparse concept tokens as the actual latent representation of the image, which are learned to jointly support pixel-level reconstruction and high-level semantic understanding, serving as a unified representation.
>
> [1] Semantic-guided masking for learning masked autoencoders. [2] Beit: Bert pre-training of image transformers.  [3] Beit v2: Masked image modeling with vector-quantized visual tokenizers.

---

> > ### Author Response · Authors · 2025-11-23
> > **Authors’ Response to Reviewer A18a - 2**
> >
> > 3. "The comparison in Table 2 is not reliable. Some baselines are reported far below than the results in their original papers - e.g., according to Table 1 in the iBOT paper, linear probing accuracies for DINO and iBOT exceed 80.0%, surpassing all linear probing results in the Table 1 in the authors' paper - Note that the epochs reported in the iBOT paper are effective epochs that account for multi-crop, not the actual pre-training epochs used by iBOT or DINO. They actually pre-trained only for 300 or 400 epochs. - Moreover, the proposed method employ MAE model parameters for initialization, which is not fair with other baselines. Also, the total epoch should be regarded as 1600 (MAE pre-training epochs) + 150/100/50 (STELLA post-training epochs) = 1750 / 1700 / 1650."
> > - Thank you for raising these important points. We agree that evaluation differences across the literature can lead to mismatches, and in our experiments all baseline models and STELLAR were evaluated under a single, consistent linear probing protocol, implemented using a unified codebase, with evaluation protocol clearly described in the revised manuscript (Section 5 and Appendix C). This avoids confounding effects from different data augmentations, pooling rules, normalization choices, or training schedules used across original papers, providing apples-to-apples comparisons under one controlled setup, as is standard in many SSL works. To ensure full reproducibility, we will open-source link to the full training and evaluation code in the camera-ready version.
> > - **Pooling choice for iBOT** We appreciate this observation. We found the gap is due to the token pooling choice in linear probing.
> >   - We stated our unified pooling choice in the revision: _As the SSL methods varies across different baseline models, for classification tasks we used the mean-pooled feature from the representations where the corresponding SSL method was performed, e.g. the global CLS token for DINO, and dense patch tokens for MAE._
> >   - We discovered that iBOT is highly sensitive to the pooling strategy. Although iBOT learns both CLS and dense patch tokens, it highly relies on the CLS token to capture global semantics. When using patch tokens (the SSL tokens used for its MIM objective), accuracy drops by 5–6 points. Using mean pooling across all tokens also reduced accuracy.
> >   - To make this transparent, we added Table 6 comparing different pooling strategies across baselines.
> >
> > |                | DINO                |                        | MAE                 |                         | iBOT                      |                         |                      | STELLAR (ours)      |                         |
> > |----------------|---------------------|------------------------|---------------------|--------------------------|----------------------------|--------------------------|----------------------|----------------------|-------------------------|
> > | **tokens**     | **global**          | dense                  | global              | **dense**                | global                     | dense                    | **gl.+de.**          | **sparse**           | dense                   |
> > | **lin. acc.**  | **76.46**           | 70.31                  | 65.61               | **66.32**                | **76.40**                  | 71.44                    | 71.58                | **73.26**            | 72.21                  |
> >
> >  - For most SSL models, the best linear probing features are the ones directly optimized by their SSL objective, such as MAE and DINO. iBOT is a notable exception, as CLS performs substantially better than its patch tokens. This explains the reviewer’s observed discrepancy.
> >
> > 4. "A lot of recent self-supervised learning methods are missing.
> > - e.g., the references below [1-22]
> > - The proposed method should also be compared with these methods"
> >
> > - We appreciate the detailed list of references. In the revised manuscript, we have expanded the set of baselines and now include:
> >   - data2vec
> >   - MoCo v3
> >   - SiameseIM
> >   - SemMAE
> >   - MSN (already included in the original version)
> > - Together with the _12_ previous baselines, we now benchmark a total of _16_ SSL methods with different model sizes, covering contrastive learning, masked image modeling, augmentation alignment, and hybrid frameworks. A complete list is provided in Table 7, sorted by method family with references for clarity.

---

> ### Author Response · Authors · 2025-11-23
> **Authors’ Response to Reviewer A18a - 3**
>
> 5. "Some core evaluation tasks are missing - e.g., Detection and segmentation on COCO"
> - STELLAR is designed to learn sparse latent representations with explicit semantic-spatial factorization, and our evaluation focuses on the standard benchmarks used for assessing general-purpose self-supervised representations under this regime, e.g. linear probing for classification and semantic segmentation. These tasks evaluate global and spatial discriminability without requiring a task-specific detection heads.
> - COCO detection typically requires architectures tailored for dense prediction (e.g., FPNs, ViTDet, Mask2Former, multi-scale features), which STELLAR does not aim to replace. Additional architectural components are beyond the scope of this work. Therefore, to evaluate spatial expressiveness fairly, we instead include linear probing on segmentation, which is widely used as a proxy for dense spatial semantics in SSL frameworks. We believe this aligns more closely with the goals and structure of STELLAR as a representation learning method.
>
> 6. “Important comparison results are missing - Fine-tuning performance comparison is very important in self-supervised learning area on visual data. However, I'm also suspecting that the proposed design may improve only linear-probing performance rather than fine-tuning performance. This concern is amplified by the utilization of the MAE initialization since MAE is well-known to show strong fine-tuning performance and weak linear-probing performance.”
> - Thank you for this suggestion. We have now conducted full fine-tuning experiments for both ImageNet-1K classification and ADE20K semantic segmentation, with complete implementation details provided in Appendix C.4 of the revised manuscript. Across these downstream tasks, STELLAR achieves best or near-best performance relative to strong SSL baselines, confirming that the learned representation also provides a competitive initialization for end-to-end adaptation. These experiments show that STELLAR is not limited to improving linear probing performance, but also consistently transfers to full fine-tuning scenario.
> - While our primary objective is to learn frozen representations capable of supporting both pixel-level reconstruction and high-level semantics, we agree that fine-tuning results offer an important complementary perspective. For completeness, we include these results in the supplementary material. We believe they further demonstrate the applicability and robustness of the proposed representation beyond linear probing.
>
> | Model | ImageNet-1K Acc. | ADE20K mIoU |
> | ------------------ | ----------------- | ------------------ |
> | **DINO** | 79.58 (+3.12) | 39.22 (+12.35) |
> | **MAE** | 77.75 (+11.43) | 40.33 (+9.42) |
> | **iBOT** | 80.72 (+9.14) | 42.76 (+10.97) |
> | **STELLAR (ours)** | 80.05 (+6.78) | 41.98 (+10.65) |
>
> We thank the reviewer for the detailed comments and constructive suggestions. We have added the requested clarifications, expanded the experiments, and revised the manuscript accordingly (with edits highlighted in blue). We hope that the improvements made in response to the reviewer’s feedback help convey the contribution of the work more clearly. If the revisions resolve the reviewer’s concerns, we would sincerely appreciate a reconsideration of the evaluation score. Thank you again for your time and thoughtful review.

---

### Official Review · Reviewer_VRTX · 2025-11-08

**Soundness:** 2
**Presentation:** 3
**Contribution:** 2
**Rating:** 4
**Confidence:** 4

**Summary:**

This paper proposes STELLAR, a self-supervised learning framework designed to learn sparse visual representations from images alone. The core idea of STELLAR is based on a low-rank matrix factorization, $V = LS$. The STELLAR framework is trained using a joint objective function that includes a reconstruction loss, a sparse concept clustering loss, a set alignment loss, and a KoLeo regularization term.The authors claim that this method, using as few as 8 latent tokens, can produce a single representation that simultaneously support high-quality image understanding, detailed pixel-level reconstruction, and fine-grained semantic understanding.

**Strengths:**

1. The paper tries to address the important problem of redundancy in dense visual representations. The motivation to learn a single, unified representation that excels at both high-level semantic understanding and low-level reconstruction is a valuable.

**Weaknesses:**

1. The paper's claims of novelty are further undermined by a profound misrepresentation of its core mechanism. The authors claim to "parameterize both S and L as learnable variables" 1 as part of a "low-rank matrix factorization".1These two statements are mutually exclusive. $L$ cannot simultaneously be a set of learnable parameters and a computed output. This formulation is not matrix factorization in the algebraic sense (like NMF or SVD).The paper's actual mechanism is a standard attention operation, cloaked in the language of classical optimization. $S \in \mathbb{R}^{r \times d}$ is a set of $r$ learnable "latent query vectors". $U \in \mathbb{R}^{n \times d}$ is the dense patch-level feature map from the ViT encoder. Equation 5 is a standard cross-attention operation, where $S$ acts as the query  and $U$ acts as the key. $L$ is simply the resulting $n \times r$ attention map, normalized via softmax.The "reconstruction" $V=LS$ 1 is then just an attention-pooled representation, where $S$ (the semantic concepts) are the values (V).Therefore, calling this "low-rank convex semi-nonnegative matrix factorization" is a profound misrepresentation of a standard attention mechanism. This attempts to invent novelty where none exists.
2.  The method proposed in this paper is actually very similar to TokenLearner [1], only with different presentation. At the same time, the paper keeps claiming to learn a sparse visual representation, but in practice, it still **relies on a standard visual encoder to extract dense visual feature**. A truly sparse architecture should employ a flexible backbone that can adaptively extract visual features.

[1] TokenLearner: What Can 8 Learned Tokens Do for Images and Videos? NeurIPS 2021

**Questions:**

1. In Appendix A.4, the manuscript states: "For efficient training, we initialized the model from public MAE checkpoint". This is a fatal confounder. The paper is presented as a novel self-supervised learning method that learns representations from scratch. But it is a fine-tuning procedure for a different existing model (MAE) actually.

---

> ### Author Response · Authors · 2025-11-22
> **Authors’ Response to Reviewer VRTX**
>
> We thank the reviewer for the time spent evaluating our submission, and for recognizing the importance of learning unified sparse visual representations that support both semantic understanding and pixel-level reconstruction. Some of the comments indicate that certain aspects of the formulation and contributions may not have been sufficiently emphasized or clearly communicated in the original version. We appreciate the opportunity to clarify these points, and we have revised the manuscript to more explicitly highlight the conceptual distinctions and methodological contributions of STELLAR. Our detailed responses are provided below.
>
> **Q1**: "The paper's claims of novelty are further undermined by a profound misrepresentation of its core mechanism. The authors claim to "parameterize both S and L as learnable variables" 1 as part of a "low-rank matrix factorization".1These two statements are mutually exclusive. $L$ cannot simultaneously be a set of learnable parameters and a computed output. This formulation is not matrix factorization in the algebraic sense (like NMF or SVD).The paper's actual mechanism is a standard attention operation, cloaked in the language of classical optimization. $S \in \mathbb{R}^{r \times d}$ is a set of $r$ learnable "latent query vectors". $U \in \mathbb{R}^{n \times d}$ is the dense patch-level feature map from the ViT encoder. Equation 5 is a standard cross-attention operation, where $S$ acts as the query and $U$ acts as the key. $L$ is simply the resulting $n \times r$ attention map, normalized via softmax.The "reconstruction" $V=LS$ 1 is then just an attention-pooled representation, where $S$ (the semantic concepts) are the values (V).Therefore, calling this "low-rank convex semi-nonnegative matrix factorization" is a profound misrepresentation of a standard attention mechanism. This attempts to invent novelty where none exists."
> - We appreciate the reviewer’s detailed feedback and agree that the original exposition could have more clearly distinguished our formulation from classical matrix factorization and from standard attention mechanisms. We have revised the manuscript accordingly to improve clarity, with edits highlighted in blue.
> - **Clarifying the use of “low-rank” and the role of S and L**
>   - Our method does not perform matrix factorization in the algorithmic sense (e.g., NMF, SVD), and we have now made this more explicit in the revised text. As stated in the _original_ submission, we do not apply NMF to encoder features. Instead, **S** and **L** are learnable latent variables produced by the encoder, and the product **V=LS** serves as a low-rank form of the dense feature map. The constraint on **L** (nonnegativity and simplex normalization) allows this form to be viewed through the lens of convex semi-NMF, but our use of the term is conceptual rather than algorithmic.
>   - We have revised the manuscript to articulate this interpretation more clearly: _Compared to a canonical dense representation of shape n × d, V = LS can be considered as a form of low-rank matrix approximation from the sparse representation. Critically, we enforce that this low-rank approximation can reconstruct the original image through some decoder D: D(LS) ≈ X.
> While the form in equation 2 resembles the low-rank structure used in convex semi-nonnegative matrix factorization (Ding et al., 2008), we do not perform NMF or any matrix factorization algorithm on the feature map (i.e. LS ≈ E(X)). Instead, both S and L are learnable latent variables output directly from the forward pass of the encoder, and their product is decoded back to the original image (D(LS) ≈ X), allowing an autoencoder-style training._
>   - **Intent behind the NMF analogy** Our goal in referencing convex semi-NMF was not to claim algorithmic novelty in matrix factorization, but to provide intuition for the semantic-spatial decomposition. The analogy helps convey why the factorized form **LS** is expressive and interpretable. We have revised the relevant text to avoid potential overinterpretation of this connection.
> - **Relationship to attention**
>   - We fully acknowledge that the computation of **L** resembles the attention weights of a single-head cross-attention layer, and we now explicitly state this in the revised manuscript: _This mapping is structurally similar to the attention weights obtained in a single-head cross-attention layer, up to the use of L2 normalization and an explicit temperature parameter. We adopt this simple formulation to compute the L matrix, and found it to be stable and effective for learning sparse concept localization across all experiments._
>   - Crucially, the proposed framework is a latent representation in the form of semantic-spatial factorization **(S ,L)**, not the mechanism that computes **L**. The distinction matters because we use this factorization as the central _inductive bias_, and jointly train it with reconstruction, clustering, and set alignment to produce unified representation.

---

> > ### Author Response · Authors · 2025-11-23
> > **Authors’ Response to Reviewer VRTX (Q2)**
> >
> > **Q2**: "The method proposed in this paper is actually very similar to TokenLearner [1], only with different presentation. At the same time, the paper keeps claiming to learn a sparse visual representation, but in practice, it still relies on a standard visual encoder to extract dense visual feature. A truly sparse architecture should employ a flexible backbone that can adaptively extract visual features."
> > - We appreciate the reviewer’s comments and the opportunity to clarify how STELLAR differs from TokenLearner and from prior work that uses dense ViT encoders.
> > - **STELLAR does not claim architectural novelty in the encoder.** Our method indeed uses a standard ViT encoder to produce initial patch features. The novelty lies not in redesigning the backbone, but in the latent representation formulation of a semantic-spatial factorization and trains this factorized representation to simultaneously support pixel-level reconstruction, fine-grained semantics, and global understanding. The sparse tokens are the final latent representation, not an architectural efficiency mechanism.
> > - To avoid confusion, we revised the text in the manuscript: _We note that the framework is flexible and does not prescribe any specific encoder or decoder architecture. In this work, we adopt a simple design with common modules to obtain S and L_
> > - To avoid any ambiguity, we added a dedicated **What is not new?** paragraph in Section 3 clarifying that:
> >   - We purposely build on widely used components rather than proposing a new encoder.
> >   - Learnable latent queries have been used in prior architectural contexts (e.g., Sparse R-CNN, TiTok).
> >   - The cosine-similarity–softmax mapping is a standard scoring operation that appears in many prior methods, e.g. attention, contrastive learning InfoNCE, etc
> >   - _**What is not new?** As our focus is not on architectural innovation, we deliberately build on simple, widely used components. The use of learnable latent queries has appeared in Sparse R-CNN (Sun et al., 2021), TiTok (Yu et al., 2024), and many others. The cosine-similarity–softmax mapping is a standard scoring operation that appears in many contexts, including single-head attention (Vaswani, 2017), contrastive learning and InfoNCE objectives (Oord et al., 2018; Chen et al., 2020a)._
> > - **Distinction from TokenLearner.**
> >   - We also expanded the Related Work section to directly address TokenLearner. TokenLearner integrates a sparse token selection module into ViT to reduce token count and improve efficiency during supervised training. Its tokens are still intertwined with spatial information and are not designed to serve as a unified representation across reconstruction and understanding.
> >   - In contrast, STELLAR:
> >     - treats sparse tokens as the latent representation, not a token-reduction mechanism in model architecture;
> >     - learns them in a fully self-supervised manner, without labels or image–text supervision
> >     - optimizes them through reconstruction, clustering, and optimal-transport set alignment, shaping them into transferable semantic concepts;
> >     - enables the same set of sparse tokens to support semantic understanding and pixel-level reconstruction.
> >     -These goals and mechanisms differ fundamentally from TokenLearner.
> > - **On the use of dense encoders.** The reviewer notes that a “truly sparse architecture” would require a different backbone. While this is an interesting direction, it is orthogonal to our contribution. Our aim is to learn a sparse latent representation, not a sparse encoder architecture. As we show, a standard encoder is sufficient for STELLAR to learn a compact and expressive sparse representation that outperforms prior sparse-token methods and matches (or exceeds) dense SSL baselines across tasks.

---

> ### Author Response · Authors · 2025-11-23
> **Authors’ Response to Reviewer VRTX (Q3)**
>
> **Q3**: "In Appendix A.4, the manuscript states: "For efficient training, we initialized the model from public MAE checkpoint". This is a fatal confounder. The paper is presented as a novel self-supervised learning method that learns representations from scratch. But it is a fine-tuning procedure for a different existing model (MAE) actually."
> - Thank you for raising this concern. We agree that the training description could have been clearer, and we have substantially revised it (with edits shown in blue). Below we clarify the training setup and how STELLAR relates to MAE.
> - We initialize only the ViT backbone weights from a publicly available MAE checkpoint. All STELLAR-specific components, including the sparse latent queries, the spatial projection layers used to compute L, the decoder, the prototype space, and all SSL objectives, are trained entirely from scratch.
> - This is not a fine-tuning procedure for MAE. The learned representation is structurally different: STELLAR produces a factorized sparse representation, whereas MAE produces dense patch embeddings.
> - It’s common in the SSL literature to use another pretrained model, e.g. BEiT [1] used DALL-E as the image reconstruction target, SemMAE[2] used iBOT to guide the training, and BEiT v2 [3] used DINO and CLIP as the teacher model for semantic reconstruction. In our work, MAE initialization provides no additional data, labels, or supervision, and does not introduce any semantic structure into STELLAR beyond what is already present in ImageNet-1K.
> - Ablations demonstrating non-dependence on MAE: In Table 5 of the revised manuscript, we provide ablations of different initialization and training strategies:
>   - Using MAE or DINO initialization for the ViT backbone leads to similar reconstruction and global semantic performance; MAE yields somewhat better fine-grained segmentation.
>   - When training all modules from scratch, STELLAR still learns meaningful sparse tokens. With a momentum-encoder warm-up of 150 epochs followed by standard training (75 epochs), reconstruction and segmentation performance catch up to the SSL-initialized backbone, while global semantic performance remains lower.
>   - These results demonstrate that MAE initialization is a practical acceleration, not a conceptual requirement.
> - Why STELLAR is not “fine-tuning MAE”: The core contribution of STELLAR is the semantic-spatial factorization and the SSL objectives that shape sparse concept tokens to jointly support pixel-level reconstruction and semantic understanding. These properties do not come from MAE: the sparse latent queries, localization mechanism, clustering, and OT alignment are all learned newly. The resulting representation is qualitatively and structurally different from MAE’s dense patch embeddings.
> - We have clarified these points in the revised manuscript to avoid any misinterpretation. We hope this resolves the concern that MAE initialization is a confounding factor and illustrates that STELLAR is not fine-tuning MAE but rather learning a new sparse representation with a practical warm-start for the backbone.
>
> We thank the reviewer once again for the time and effort invested in evaluating our submission. We hope that the clarifications, additional experiments, and revisions provided here address the concerns raised and offer a more accurate understanding of the contributions and novelty of STELLAR. We respectfully request that the reviewer reconsider the assessment in light of these clarifications.
>
> [1] Bao, H., Dong, L., Piao, S., & Wei, F. (2021). Beit: Bert pre-training of image transformers. arXiv preprint arXiv:2106.08254.
> [2] Li, G., Zheng, H., Liu, D., Wang, C., Su, B., & Zheng, C. (2022). Semmae: Semantic-guided masking for learning masked autoencoders. Advances in Neural Information Processing Systems, 35, 14290-14302.
> [3] Peng, Z., Dong, L., Bao, H., Ye, Q., & Wei, F. (2022). Beit v2: Masked image modeling with vector-quantized visual tokenizers. arXiv preprint arXiv:2208.06366.

---

> ### Comment · Reviewer_VRTX · 2025-11-26
> **Response to Authors**
>
> Thank you for the detailed response. While the clarifications are helpful, I still have several concerns that are not fully resolved.
>
> 1. In the rebuttal, the authors explicitly position STELLAR as a component-level method that builds on standard ViT backbones, standard cosine-similarity–softmax scoring. Given this, it is not obvious that it should be described as a general “sparse vision representation framework”. The work seems closer to a particular way of post-processing or refactoring features from an existing encoder into a factorized representation, rather than proposing a broadly applicable framework for sparse vision representations. I would encourage the authors to torn down claims.
>
>
> 2. I agree that architectural novelty is not a prerequisite for a good paper. However, in that case the empirical evidence needs to be particularly convincing. Here, the results in Table 5 actually reinforce my concern that the method is heavily dependent on strong SSL initialization and that the proposed factorization discards useful information. When moving away from MAE/DINO initialization to the “EMA only” and “EMA+75ep.” settings, there is a substantial drop in performance (e.g., Class. Acc. and Seg. mIOU both degrade noticeably). This suggests that the framework, when trained truly from scratch, struggles to retain the level of semantic information that the dense SSL backbones provide. Moreover, the table also omits the Reconstruction FID for the original MAE and DINO baselines. From prior work, MAE can often achieve much better rFID than what is reported for STELLAR in Table 5. Without reporting the MAE/DINO rFID under the same evaluation protocol, it is difficult to judge whether the proposed factorization preserves or in fact loses fine-grained visual detail relative to the underlying backbone. These points make it hard to accept the claim that STELLAR offers a clear “sparse visual representation” rather than a lossy refactoring of an already strong dense SSL model.

---

> > ### Author Response · Authors · 2025-11-27
> > **Follow-up Response to Reviewer VRTX (Q1)**
> >
> > We thank the reviewer for the additional time and attention given to our submission. We are glad that the earlier clarifications were helpful, and we appreciate the opportunity to further address the remaining concerns. Upon careful reading, several of the points raised appear to stem from misunderstandings about the nature of the proposed representation and the training setup. We provide precise clarifications below to help accurately assessment the contribution of this work.
> >
> > 1. “the authors explicitly position STELLAR as a _component-level_ method that builds on standard ViT backbones, standard cosine-similarity–softmax scoring”
> >
> > **Response**: We did not position STELLAR as a “component-level method” in the manuscript or rebuttal. In the revised submission, we explicitly formalized the contribution as a representation framework, defined by:
> > - **latent representation form**: $S, L = \mathcal{E}(X)$, where $S$ encodes semantic concepts and $L$ encodes their spatial distributions (line 172),
> > - **reconstruction requirement**: $\mathcal{D}(L S) \approx X$, ensuring that the sparse tokens retain fine-grained pixel information (line 173),
> > - **semantic learning objectives**: which operate directly on the sparse latent variables, not on dense features (Sections 3.2–3.3).
> >
> > This structure is the basis for calling STELLAR a sparse vision representation framework, rather than a particular architectural component.
> >
> > To avoid overstating novelty, we’ve already made explicit in the revised manuscript that the framework does _not_ rely on custom architectural modules:
> >
> > - _“The framework is flexible and does not prescribe any specific encoder or decoder architecture. In this work, we adopt a simple design with common modules to obtain Sand L.”_ (line 174)
> >
> > - And we added a dedicated clarification: _“**What is not new?** As our focus is not on architectural innovation, we deliberately build on simple, widely used components.”_ (line 278)
> >
> > Thus, the novelty of STELLAR lies in the representation form and the learning formulation, not in any single architectural component.
> >
> > 2. “The work seems closer to a particular way of post-processing or refactoring features from an existing encoder into a factorized representation, rather than proposing a broadly applicable framework for sparse vision representations”
> >
> > **Response**: We have further clarified in the revision:
> > - _“we do not perform NMF or any matrix factorization algorithm on the feature map (i.e. LS ≈ E(X)). Instead, both S and L are learnable latent variables output directly from the forward pass of the encoder, and their product is decoded back to the original image (D(LS) ≈ X), allowing an autoencoder-style training.”_ (line 165)
> >
> > To avoid confusion, we clarify what we mean by “post-processing” in the context of representation learning.
> >
> > A post-processing method typically:
> > - Takes dense encoder features as fixed input
> > - Computes (rather than learns) a new representation through a non-learned transformation (e.g., clustering, NMF, token selection)
> > - Does not update the encoder weights, since the sparse representation is not part of the end-to-end learning process.
> >
> > In contrast, STELLAR:
> > - Directly encodes the image into the latent variables S and L, which cannot be written as a function of dense ViT features alone,
> > - Learns $S, L = \mathcal{E}(X)$ jointly through reconstruction, clustering, and set-alignment objectives
> > - Optimizes the entire encoder end-to-end, meaning the sparse representation is learned, not derived.
> >
> > Based on these, as well as the clarification to the previous point, the proposed method is a generic framework for sparse vision representation.

---

> ### Author Response · Authors · 2025-11-27
> **Follow-up Response to Reviewer VRTX (Q2)**
>
> 3. “Table 5 actually reinforce my concern that the method is heavily dependent on strong SSL initialization and that the proposed factorization discards useful information. When moving away from MAE/DINO initialization to the “EMA only” and “EMA+75ep.” settings, there is a substantial drop in performance (e.g., Class. Acc. and Seg. mIOU both degrade noticeably). This suggests that the framework, when trained truly from scratch, struggles to retain the level of semantic information that the dense SSL backbones provide.”
>
> **Response**: The revised manuscript adds substantial experiments addressing this point, and the evidence does not indicate that STELLAR “discards useful information.” Rather, the ablation clarifies the role of initialization.
> - STELLAR significantly improves over MAE even when _only_ the ViT blocks are warm-started
>   - In all settings, every STELLAR-specific component (latent queries, projection layers, decoder, prototypes, clustering heads) are trained from scratch.
>   - With MAE initialization in the ViT blocks, STELLAR achieves:
>     - 73.26% classification linear probing (vs. MAE 66.32% under identical evaluation)
>     - 31.33% mIoU segmentation linear probing (vs. MAE 30.91%)
>     - 3.14 reconstruction FID with 16 sparse tokens (vs. MAE reconstruction with 16 tokens 150.73 FID)
>   - These results illustrate that the STELLAR factorization does not discard information. Instead, it substantially enhances both semantic and fine-grained detail relative to the initialization backbone.
>
> - Training entirely from scratch is feasible with standard SSL practice. With 150 EMA warm-up epochs + 75 standard epochs (225 total), the model reaches:
>   - Reconstruction: 3.21 rFID (matches MAE/DINO initialization: 3.14 / 3.31)
>   - Segmentation: 28.10 mIoU (matches DINO init: 28.17; close to MAE init: 31.33)
>   - Classification: 65.28% (higher than nearly half the baselines, and notably comparable to models such as MAE (66.32% with 800 pretraining epochs) and AIM (2 billion pretraining data, 63.78%-66.86% with significantly larger model size at 600M-1B))
>
> These results show that STELLAR does learn meaningful sparse representations from scratch under standard SSL training budgets. The differences between scratch vs MAE initialization are expected, not evidence of dependency
> - MAE provide 800–1600 epochs of distilled invariances.
> - STELLAR from scratch uses a much smaller compute budget (225 epochs).
> - Learning a unified sparse representation with reconstruction + semantics + clustering + OT alignment is a multi‐component problem where warm-starting the backbone improves optimization stability.
>
> Thus, MAE initialization is a practical stabilizer, not a conceptual requirement. Crucially, it introduces no additional data or supervision beyond, and is easily applicable. We hope this clarifies the intent of the ablation table and resolves the concern regarding dependency on MAE.
>
> 4. “The table omits the Reconstruction FID for MAE and DINO… it is difficult to judge whether the proposed factorization preserves or in fact loses fine-grained visual detail relative to the underlying backbone.”
>
> **Response**:
> - DINO is not a reconstruction-based method, so rFID is not defined for DINO under standard evaluation protocol.
> - MAE rFID is already reported in Table 1 of the paper. Using the original MAE encoder-decoder, we obtain:
>   - 150.73 rFID with 16 tokens
>   - 131.01 rFID with 32 tokens
>
>   These results are substantially worse than STELLAR, which achieves:
>   - 3.14 rFID (16 sparse tokens)
>   - 3.68 rFID (8 sparse tokens)
>   - 2.60 rFID (16 tokens, ViT-H backbone)
> - We also provided qualitative comparisons in Figure 3 showing that MAE reconstructions are characteristically blurry, while STELLAR preserves fine structures even with only 16 tokens.
> - Importantly, STELLAR simultaneously achieves strong reconstruction and semantic performance with sparse representation (e.g., 79.10% IN-1K linear probing, 2.6 FID).
> - We note that a dense reconstruction model with 256 tokens such as Taming-VQGAN [1] achieved 7.94 rFID. STELLAR is already approaching the limit with only 8-16 tokens.
>
> Clearly, the theoretical limit of information in 16 sparse tokens is definitely way smaller than hundreds of tokens in a standard dense representation. Therefore it is challenging to preserve both high-level semantics and low-level reconstruction details about the image. However, we show that it is possible if the form of representation properly disentangles semantic and spatial information, and provided training recipes to achieve this practically.
>
> We hope these clarifications resolve the concerns and enable a fairer evaluation of the contribution.
>
>
> [1] Esser, P., Rombach, R., & Ommer, B. (2021). Taming transformers for high-resolution image synthesis. In Proceedings of the IEEE/CVF conference on computer vision and pattern recognition (pp. 12873-12883).

---

### Author Response · Authors · 2025-11-23

We thank all reviewers for the time and thoughtful feedback provided during the evaluation of our submission. Both positive and critical comments helped us substantially strengthen the clarity, completeness, and presentation of the work. Below, we summarize the key improvements and clarifications in the revised submission, which we hope convey our contributions more accurately.

**Strengths highlighted by reviewers**
- The motivation for learning sparse visual representations is clear and timely.
- Separating _“what”_ (semantic concepts) from _“where”_ (spatial information) is seen as a compelling modeling idea.
- The method is well-motivated, generally well-written, and empirically strong across reconstruction, linear probing, and segmentation.
- Sparse tokens are recognized as promising for interpretability, efficiency, and representation quality.

We appreciate these encouraging remarks, which reflect the intent of the work. We also took the feedbacks in the revision by expanding the following aspects.

**Clarification of core contribution**
- Several questions stemmed from incomplete or ambiguous explanations in the original submission. The revised version (with edits highlighted in blue) now clarifies:
  - The conceptual framing of STELLAR as a sparse latent vision representation that models an image using a small number of tokens by semantic-spatial factorization
  - How the proposed training scheme learns a unified representation that simultaneously supports pixel-level reconstruction (FID 2.60) and semantic understanding (79.10% IN-1K linear probing), using as few as 8–16 sparse tokens;
  - the distinction between STELLAR’s factorized sparse representation and classical NMF, attention mechanisms, TokenLearner, and SemMAE
- To avoid ambiguity, we also added a dedicated **“What’s not new?”** paragraph explicitly stating the components we do not claim as our novelty (e.g., model architecture), which were included for completeness.
- Clarification of the proposed framework in contrast to "component-level method" and "post-processing", based on the follow-up comments from Reviewer VRTX.

**Expanded experiments**
- Expanded baselines: four additional SSL methods were added in addition to the previous 12 baselines, bringing the total to 16 representative SSL approaches across contrastive learning, alignment-based SSL, masked image modeling, hybrid methods, and sparse representation frameworks (Table 1, 2, 7).
- Ablation experiment showing the critical effect of the concept clustering loss (Table 4)
- Initialization ablations (MAE, DINO, training from scratch with EMA warm-up) demonstrating that MAE initialization is beneficial to training but is not a dependency for the STELLAR framework (Table 5).
- Pooling-sensitivity analysis (Table 6) showing how token-pooling choices affect linear probing performance across models, explaining discrepancies with results reported in other papers.
- Fine-tuning experiments for ImageNet-1K classification and ADE20K segmentation (Table 11), where STELLAR achieves best or near-best performance.
- Computational cost analysis, including time and memory for clustering and optimal transport matching, and comparisons between Sinkhorn and Hungarian matching algorithms (Table 12).

We sincerely appreciate the reviewers’ careful reading and constructive feedback. The revised manuscript incorporates all feasible suggestions and addresses misunderstandings raised during the review process. We hope the expanded analysis, added experiments, and clearer explanations help convey the contribution of STELLAR. We would be grateful if the improvements are reflected in the final evaluation.

---

### Meta-Review · Area_Chair_8WD7 · 2026-01-03

**Summary:**

The reviewers acknowledged the motivation of the paper and the experiments provided. However, they also raised concerns in the following aspects:
1. Contribution and novelty of the work is limited as there are already a lot of similar work.
2. Missing baseline and result discussions.
3. Several questions regarding the details of the implementation and experiments.

The authors addressed the concerns by providing more discussions on novelty and comparison during rebuttal.

**Reviewer Concerns:**

What's addressed:
1. comparisons with baseline models/methods such as DINO, MAE, iBOT, etc.
2. The motivation of STELLAR and its comparison and main differentiating factor with existing methods.

What's not addressed:
The novelty seems still quite limited.

**Reviewer Scores:**

The reviewers will likely keep their scores.

---

### Decision · Program_Chairs · 2026-01-26

Reject